# In Silico Approach for the Evaluation of the Potential Antiviral Activity of Extra Virgin Olive Oil (EVOO) Bioactive Constituents Oleuropein and Oleocanthal on Spike Therapeutic Drug Target of SARS-CoV-2

**DOI:** 10.3390/molecules27217572

**Published:** 2022-11-04

**Authors:** Elena G. Geromichalou, George D. Geromichalos

**Affiliations:** 1Laboratory of Pharmacology, Medical School, National and Kapodistrian University of Athens, 75 Mikras Asias Street, 11527 Athens, Greece; 2Department of General and Inorganic Chemistry, Faculty of Chemistry, Aristotle University of Thessaloniki, 54124 Thessaloniki, Greece

**Keywords:** extra virgin olive oil (EVOO) constituents, molecular docking, SARS-CoV-2 Spike protein, SARS-CoV-2 mutation variants, omicron-delta-alpha-beta-gamma-epsilon-kappa

## Abstract

Since there is an urgent need for novel treatments to combat the current coronavirus disease 2019 (COVID-19) pandemic, in silico molecular docking studies were implemented as an attempt to explore the ability of selected bioactive constituents of extra virgin olive oil (EVOO) to act as potent SARS-CoV-2 (severe acute respiratory syndrome coronavirus 2) antiviral compounds, aiming to explore their ability to interact with SARS-CoV-2 Spike key therapeutic target protein. Our results suggest that EVOO constituents display substantial capacity for binding and interfering with Spike (S) protein, both wild-type and mutant, via the receptor-binding domain (RBD) of Spike, or other binding targets such as angiotensin-converting enzyme 2 (ACE2) or the RBD-ACE2 protein complex, inhibiting the interaction of the virus with host cells. This in silico study provides useful insights for the understanding of the mechanism of action of the studied compounds at a molecular level. From the present study, it could be suggested that the studied active phytochemicals could potentially inhibit the Spike protein, contributing thus to the understanding of the role that they can play in future drug designing and the development of anti-COVID-19 therapeutics.

## 1. Introduction

In the last decade, there has been an increase in the acceptance of herbal treatment [1]. Natural products have long been used in traditional medicines to treat various diseases, and purified phytochemicals from medicinal plants provide a valuable scaffold for the discovery of new drug leads. Furthermore, natural products have proven to be safe and easily available to treat coronavirus-affected patients. A great number of phytochemicals have been identified to interact with severe acute respiratory syndrome coronavirus 2 (SARS-CoV-2), acting as COVID-19 potential therapeutics [2,3,4,5,6,7,8,9,10]. Several potential therapeutic approaches have been experimented to treat SARS-CoV-2 infection, such as protein-based vaccine design, the blocking of the angiotensin-converting enzyme 2 (ACE2) receptor, and the effect of phytochemicals on spike protein binding with its ACE2 receptor. Bioactive phytochemicals’ antiviral activity on SARS-CoV-2 was investigated with the aid of molecular docking [11,12,13,14,15,16,17,18,19,20,21,22,23,24,25,26,27,28,29]. Furthermore, a combined approach of virtual drug screening, molecular docking, and supervised machine learning techniques were employed in order to identify candidate drugs of chemical libraries with natural products with potential antiviral activity on SARS-CoV-2 [30].

*Olea europaea* L. is the most well-known plant in the Olea genus [31]. The polyphenols generated by the olive tree (Olea europaea) are found mainly in the tree’s leaves and drupes. Although the health benefits of extra virgin olive oil (EVOO) and olive leaf extracts have long been recognized, they have only lately been thoroughly investigated. The total extract of olive leaves and olive oil and their compounds were reported in several studies for their antiviral, anti-inflammatory, immunomodulatory, anticancer, anti-Alzheimer’s disease, and antithrombotic activities [32,33,34,35,36,37,38,39,40,41,42,43,44]. Olive leaves were reported to be rich in phenolic compounds such as oleuropein, hydroxytyrosol, verbascoside, apigenin-7-*O*-glucoside, and luteolin-7-*O*-glucoside, as well as triterpenoids such as maslinic, ursolic, and oleanolic acids that have been reported as anti-SARS-CoV-2 metabolites in recent computational and in vitro studies. Two of the important phytochemical constituents reported to be isolated and detected in the extracts of Olea europaea leaves are Oleuropein (OEU) and oleocanthal (OC). OEU, the major polyphenolic compound enriched in olive oil and leaves of the olive tree, has attracted scientific attention in recent years because of a variety of reported health benefits. Reviewing the recent research, olive leaves were selected as a potential co-therapy supplement for the treatment and improvement of clinical manifestations in COVID-19 patients. In addition, olive leaf extract was previously reported in several in vivo studies for its anti-inflammatory, analgesic, antipyretic, immunomodulatory, and antithrombotic activities, which are of great benefit in the control of associated inflammatory cytokine storm and disseminated intravascular coagulation in COVID-19 patients. The molecular structures of EVOO constituents oleuropein (OEU) and oleocanthal (OC) are depicted in Figure 1. OEU (Methyl (2*S*,3*E*,4*S*)-4-{2-[2-(3,4-dihydroxyphenyl)ethoxy]-2-oxoethyl}-3-ethylidene-2-{[(2*S*,3*R*,4*S*,5*S*,6*R*)-3,4,5-trihydroxy-6-(hydroxymethyl)oxan-2-yl]oxy}-2*H*-pyran-5-carboxylate) is a glycosylated secoiridoid, a type of phenolic bitter compound found in green olive skin, flesh, seeds, and leaves, and argan oil of the olive tree, *Olea europaea*. OEU consists of a molecule of elenolic acid linked to the orthodiphenol hydroxytyrosol by an ester bond, and to a molecule of glucose by a glycosidic bond. OC (2-(4-Hydroxyphenyl)ethyl (3*S*,4*E*)-4-formyl-3-(2-oxoethyl)hex-4-enoate) is a phenylethanoid, or a type of natural phenolic compound, found in EVOO. OC is a tyrosol ester, and its chemical structure is related to OEU. In addition, OEU and OC are the phenolic compounds that are mainly responsible for antioxidant activity [45].

In silico predictive tools play an important role, as they are rapid and cost-effective compared to the trial-and-error methods using experimental studies. In silico approaches are frequently used in current drug design to assist in the knowledge of drug–receptor interactions. By exposing the mechanism of drug–receptor interactions, computational methodologies have been devised in the literature to strongly support and facilitate the discovery of novel, more potent inhibitors [47,48].

Since there is an urgent need for novel treatments to combat the current COVID-19 pandemic that has resulted in a huge number of deaths and infected people, in silico studies were implemented regarding the associated SARS-CoV-2, as an attempt to explore the ability of the EVOO constituents to act as potent SARS-CoV-2 antiviral compounds, and to elucidate the possible mechanism of action. EVOO constituents have been reported as a promising phytotherapy or co-therapy against COVID-19 [49,50,51,52]. In the past, oleuropein has shown a potential antiviral activity against respiratory syncytial virus (RSV), a common upper-respiratory infection (URI) virus [53]. Recently, Hussain et al., performing docking experiments, docking validation, interaction analysis, and molecular dynamic simulation analysis, investigated the binding pattern of oleuropein against the main protease 3CLpro target of SARS-CoV-2 [54].

Herein, two bioactive constituents of EVOO, namely OEU and OC, have been selected as small phytochemical molecules in a molecular docking study of the Spike glycoprotein of SARS-CoV-2 with its human protein receptor ACE2.

Among the encoded proteins of the SARS-CoV-2 genome, the S protein is the most vital protein, which controls the biological processes such as viral particle attachment, fusion, and lastly entry in the host cell, and is thus considered a key therapeutic target for COVID-19, including intensive vaccine and therapeutic antibody research [55]. The coronavirus’s entry into host cells is mediated by the S protein containing the receptor-binding domain (RBD), which recognizes the target receptor, leading to the splicing of the trimeric S protein into subunits S1 and S2, facilitating membrane fusion; virus infection then occurs through endocytosis [56]. The S1 subunit contains an N-terminal domain (NTD) and the RBD, where the receptor-binding motif (RBM) is responsible for the interaction with the ACE2 receptor to gain entry into the host [57]. ACE2 is a transmembrane protein, which is considered as a receptor for Spike protein binding of novel coronavirus (SARS-CoV-2). The transmembrane Spike glycoproteins form homotrimers that protrude from the viral surface. The Spike trimeric glycoprotein, being critical for the entry of the coronaviruses, is an attractive antiviral target. Blocking the binding of SARS-CoV-2 Spike protein to host human ACE2 receptor on the human cell is the first and most promising approach for blocking cell entry and inhibiting SARS-CoV-2 infection [58].

Currently, six variants of concern (VOC)—Alpha (B.1.1.7), Beta (B.1.351), Gamma (P.1), Delta (B.1.617.2), Epsilon (B.1.427 and B.1.429 lineages) (United States), and Omicron (B.1.1.529) (origin South Africa)—and three variants of interest (VOI)—Kappa (B.1.617.1) (India), Lambda (C.35 and C.37) (Thailand ex. Egypt), and Mu (B.1.621)—are circulating in different parts of the world. Other variants with restricted circulation include Zeta (P.2) (Brazil), Eta (B.1.525) (United Kingdom), Theta (P.3) (Philippines), and Iota (B.1.526) (United States). Based on these data, it is interesting to study the ability of the EVOO constituents to bind to both the wild-type and mutated SARS-CoV-2 Spike protein.

Hindering the S/ACE2 receptor binding by neutralizing antibodies or antiviral drugs could inhibit viral replication by preventing viral entry to the host cells [59,60,61]. However, the presence of accelerating genetic variation of the S1 and RBD could be a real challenge against using this type of antiviral strategy [62].

In the present study, the selection of the two EVOO constituents, OEU and OC, to investigate their potential inhibitory activities on the binding of the S1′s unit RBD domain of the S protein of different SARS-CoV-2 variants with the human ACE2 receptor via molecular docking studies was based on their recently reported activity against SARS-CoV-2 as potential anti-SARS-CoV-2 drugs [49,63,64]. In a recent review, it was also revealed that secondary metabolites of olive oil, specifically oleanolic acid and oleuropein, could help combat COVID-19 infection by modifying the structure of SARS-CoV-2-binding proteins, thus hindering the virion’s ability to enter the host cell [65]. Furthermore, OEU along with hydroxytyrosol inhibit the fusion of viruses with cell membranes [66]. Since SARS-CoV-2 is an enveloped virus with Spike glycoproteins, OEU may inhibit its endocytosis.

The computational strategy employed in this study aspires to highlight the rationale to use EVOO bioactive constituents OEU and OC in the drug development as an anti-SARS-CoV-2 drugs lead.

## 2. Results and Discussion

### In Silico Molecular Docking Studies on SARS-CoV-2 Targets

Therapeutic strategies to block coronavirus from entering host cells by targeting Spike proteins or specific receptors on the host surface are valuable for the development of antiviral drugs. The RBD region is also a critical target for neutralizing antibodies. The RBD fragment (from amino acid residues 331–524 of the Spike protein) in SARS-CoV-2 strongly binds with human ACE2 (hACE2) receptor. Thus, this Spike protein fragment is responsible for the entry of SARS-CoV-2 in human ACE2-expressing cells. Small molecules, which can affect the binding efficiency of the Spike protein with its receptor, may act as the viral attachment inhibitor for the infection. As a result, the S protein can be considered as a target for the development of medicines in COVID-19, as well as SARS-CoV infection [67,68,69].

In silico molecular docking calculations were employed to evaluate the ability of EVOO constituents OC and OEU to bind to SARS-CoV-2-related viral infection target proteins, including: (a) the S protein in either down (closed) or up (open) conformation state, in both the wild-type (wt) and mutant (mt) S proteins; (b) the S protein in complex with the host human ACE2 receptor in both the wt and mt S proteins; and (c) the RBD domain of the S protein, either alone or in complex with the ACE2 receptor, in both the wt and mt S proteins.

In order to computationally study the potential antiviral activity of OC and OEU against various SARS-CoV-2 target proteins, in silico molecular docking studies were adopted on the following SARS-CoV-2 target proteins: (a) the three-dimensional structure of the full-length model of the Spike protein in open conformation in a model (based on the PDB ID: 6VSB) [70] developed by the Amaro lab [71], either alone or in complex with the human ACE2 receptor; (b) the S protein in closed conformation (RBD in down position, Protein Data Bank (PDB) ID: 6VXX); (c) the S protein in open conformation (one RBD-up conformation with D614G mutation, PDB ID: 7KDL); (d) only the RBD domain of the S protein with N501Y point mutation (PDB ID: 7NEG); (e) the S protein in complex with the ACE2 receptor (one RBD in open (up) conformation, PDB ID: 7KJ2); and (f) the RBD domain of the S protein in complex with the ACE2 receptor (PDB ID: 6VW1). Furthermore, docking calculations were employed to evaluate the ability of OEU and OC to interfere with the SARS-CoV-2 S protein with either the wt up (open) conformation state (PDB ID: 6VYB) of the protein or the mt Alpha 501Y.V1 (B.1.1.7) (United Kingdom–UK) (PDB ID: 8dli), Beta 501Y.V2 (B.1.351) (South Africa–SA) (PDB ID: 8dll), Gamma 209/501Y.V3, 484K.V2 (B.1.1.28 or P1) (Brazilian–BR) (PDB ID: 8dlo), and Epsilon (B.1.427 and the California B.1.429 lineages) (United States) (PDB ID: 8dlt) variants of S protein (the naming is according to the Phylogenetic Assignment of Named Global Outbreak (PANGO) lineages). Additional docking studies were performed on the Delta (B.1.617.2) (India) and Kappa (B.1.617.1) (India) mt S proteins in open (one RBD-up) (PDB IDs: 7v7o and 7v7e, respectively) conformations, as well as on the RBDs of the Delta and Kappa mt S proteins in complex with ACE2 protein (PDB IDs 7v8b and 7v87, respectively). Finally, further studies were performed on the Omicron (B.1.1.529) mt Spike subvariants in open conformation state with one RBD-up BA.1 (B.1.1.529.1) (PDB IDs 7TGW and 7QO7) and BA.2 (B.1.1.529.2) (PDB ID 7XIW), as well as in closed conformation state with all RBDs-down for Omicron subvariants BA.2.13 (PDB ID 7XNR), BA.3 (B.1.1.529.3) (PDB ID 7XIY), and BA.4 (PDB IDs 7XNQ and 7XNS). Finally, docking studies were performed with the Cryo-EM structure of the RBD domain of the S mt protein Omicron BA.3 subvariant (PDB ID 7XIZ), the S protein’s RBD domain of Omicron’s subvariants BA.1 (PDB ID 7WPB) and BA.2 (PDB IDs 7XO9 and 7ZF7) complexed with ACE2, the Omicron BA.2 subvariant Spike trimer with two and three human ACE2-bound (PDB IDs 7XO7 and 7XO8, respectively), the BA.2 subvariant of the Omicron Spike protein in complex with Fab BD55-5840 (PDB ID 7X6A), and Omicron’s BA.4-5 subvariant RBD in complex with Beta-27 Fab and C1 nanobody (PDB ID 7ZXU).

The best-scored pose of docked compounds in each target macromolecule was selected for the evaluation of binding interactions. Binding free energy (ΔG_bind_) for each pose was also computed and poses with the lowest binding free energy were selected for further visualization studies. Both OEU and OC showed good docking scores, reflecting drug-binding affinities with the studied proteins.

The computed binding energies for the best docking poses of the studied EVOO constituents on these target proteins are shown in Table 1 and Table 2. Better inhibition is usually reflected by low binding energy (the lower the energy required, the stronger and more specific the binding is).

The enantiomer structures of OEU and OC, indicating the chiral center of each one, are depicted in Appendix A, respectively. The computed binding energies were revealed to be almost the same in both enantiomer molecules of each compound.

**Table 1 molecules-27-07572-t001:** ΔG_bind_ glide extra precision (XP) binding energies (in kcal/mol) of EVOO constituents OEU and OC docked on SARS-CoV-2 Spike (S) protein (in open and closed conformation), the RBD of S and their complexes with monoclonal antibodies, in both wild-type (wt) and mutant (mt) variants.

SARS-CoV-2 Target Protein (wt and mt) (PDB Entry Code)	EVOO Constituents	
	OEU	OC
Wt full-length Spike protein open (based on 6VSB)	−54.30	−41.82
Wt full-length Spike closed (based on 6VXX)	−47.94	−41.42
Wt open Spike protein (one RBD-up) (6VYB)	−54.16	−29.30
Wt open Spike protein (two RBDs-up) (7A93)	−55.09	−37.57
Wt closed Spike protein (three RBDs-down) (6VXX)	−57.13	−29.23
D614G mt open Spike protein (one RBD-up) (7KDL)	−61.88	−31.79
Alpha mt open Spike protein (one RBD-up) (8DLI)	−52.84	−38.15
Beta mt open Spike protein (one RBD-up) (8DLL)	−54.98	−38.94
Gamma mt open Spike protein (one RBD-up) (8DLO)	−62.15	−38.97
Delta mt open Spike protein (one RBD-up) (7V7O)	−52.31	−31.47
Epsilon mt open Spike protein (one RBD-up) (8DLT)	−64.68	−38.33
Kappa mt open Spike protein (one RBD-up) (7V7E)	−55.72	−35.98
Omicron BA.1 mt open Spike protein (one RBD-up) (7TGW)	−49.54	−32.14
Omicron BA.1 mt open Spike protein (one RBD-up) (7QO7)	−52.94	−37.47
Omicron BA.2 mt open Spike protein (one RBD-up) (7XIW)	−50.29	−34.47
Omicron BA.2.13 mt closed Spike protein (all RBDs-down) (7XNR)	−49.01	−41.58
Omicron BA.3 mt closed Spike protein (all RBDs-down) (7XIY)	−54.66	−37.01
Omicron BA.4 mt closed Spike protein (all RBDs-down) (7XNQ)	−55.11	−35.05
Omicron BA.4 mt closed Spike protein (all RBDs-down) (7XNS)	−43.94	−37.44
N501Y mt RBD of Spike protein (7NEG)	−30.04	−42.47
Omicron BA.3 mt RBD of Spike protein (7XIZ)	−49.60	−35.23
N501Y mt RBD in complex with COVOX-269 Fab (7NEG)	−36.65	−39.46
Omicron BA.2 mt S protein in complex with Fab BD55-5840 (7X6A)	−45.26	−41.67
Omicron BA.4-5 mt RBD/Beta-27 Fab and C1 nanobody complex (7ZXU)	−36.00	−37.59

From Table 1 and Table 2, it is deduced that for the great majority of SARS-CoV-2 target proteins, OEU is better bound to the protein compared to OC. From Table 1, comparing the binding capacity of OEU and OC between the wt and mt variants of the open-conformation-state (one RBD-up) S protein, the following order is revealed: (higher binding capacity with lower ΔG_bind_) to (lower binding capacity with higher ΔG_bind_) Epsilon mt (8DLT) > Gamma (8DLO) > D614G mt (7KDL) > Kappa mt (7V7E) > Beta mt (8DLL) ≈ wt (6VYB) > Omicron BA.1 mt (7QO7) ≈ Alpha mt (8DLI) ≈ Delta mt (7V7O) > Omicron BA.2 mt (7XIW) for OEU, and Gamma (8DLO) ≈ Beta mt (8DLL) > Epsilon mt (8DLT) ≈ Alpha mt (8DLI) > Omicron BA.1 mt (7QO7) > Kappa mt (7V7E) > Omicron BA.2 mt (7XIW) > D614G mt (7KDL) ≈ Delta mt (7V7O) > wt (6VYB) for OC. Similarly, comparing the binding capacity of OEU and OC between the wt and Omicron mt variants of the closed-conformation-state (three RBDs-down) S protein, the following order is revealed: (higher binding capacity with lower ΔG_bind_) to (lower binding capacity with higher ΔG_bind_) Wt (6VXX) > Omicron BA.4 mt (7XNQ) > Omicron BA.3 mt (7XIY) > Omicron BA.2.13 mt (7XNR) for OEU, and Omicron BA.2.13 mt (7XNR) > Omicron BA.4 mt (7XNS) ≈ Omicron BA.3 mt (7XIY) > Wt (6VXX) for OC. On the other hand, OEU was found to be better bound to Omicron BA.3 mt RBD of S protein (7XIZ) compared to N501Y mt RBD of S protein (7NEG). The reverse was documented for OC.

**Table 2 molecules-27-07572-t002:** ΔG_bind_ glide extra precision (XP) binding energies (in kcal/mol) of EVOO constituents OEU and OC docked on the complex of both wild-type (wt) and mutant (mt) variants of SARS-CoV-2 Spike (S) and S proteins’ RBD with ACE2 host human protein.

SARS-CoV-2 Target Protein (wt and mt) (PDB Entry Code)	EVOO Constituents	
	OEU	OC
Wt open Spike protein/ACE2 complex (7KJ2)	−55.14	−32.34
Omicron BA.2 mt Spike/ACE2 complex (two ACE2-bound) (7XO7)	−41.15	−37.64
Omicron BA.2 mt Spike/ACE2 complex (three ACE2-bound) (7XO8)	−45.31	−35.14
Wt full-length S proteins’ RBD/ACE2 complex (from 6M17)	−46.03	−32.59
Wt S proteins’ RBD/ACE2 complex (6VW1)	−37.93	−29.48
Delta S proteins’ RBD/ACE2 complex (7V8B)	−37.14	−37.70
Kappa S proteins’ RBD/ACE2 complex (7V87)	−49.55	−38.09
Omicron BA.1 mt S proteins’ RBD/ACE2 complex (7WPB)	−37.18	−37.09
Omicron BA.2 mt S proteins’ RBD/ACE2 complex (7XO9)	−44.55	−39.97
Omicron BA.2 mt S proteins’ RBD/ACE2 complex (7ZF7)	−43.38	−40.10

From Table 2, it is obvious that OEU is better bound to the wt Spike protein/ACE2 complex compared to the Omicron BA.2 mt Spike protein/ACE2 complex (for both two and three ACE2-bound). The opposite is observed for OC. Better binding of OEU on the S proteins’ RBD/ACE2 complex is found for the Kappa mt variant (7V87) followed by the Omicron BA.2 mt variant (7XO9) and 7ZF7), wt variant (6VW1), Omicron BA.1 mt variant (7WPB), and Delta mt variant (7V8B). On the other hand, OC was predicted to be bound better to all of the mt S proteins’ RBD/ACE2 complexes compared to the wt variant (6VW1).

### 2.1. Docking Calculations on Full-Length Model of the SARS-CoV-2 S Protein

To explore the potential role of OEU and OC as promising antiviral agents against SARS-CoV-2 and the possibility to interfere with the full-length model of the glycosylated SARS-CoV-2 Spike protein in both closed and open states, based on the cryo-EM structures 6VXX and 6VSB, respectively, where all three RBDs are in a “down” (closed) conformation [72] and the RBD within chain A (RBD-A) is in an “up” (open) conformation, respectively [70], we employed molecular docking studies on a model developed by the Amaro Lab: PSF/PDB for the full-length Spike protein in the open and closed states, including protein, glycans, membrane, water, and ions.

#### 2.1.1. Wt Full-Length Spike Protein Open (Based on 6VSB)

The three-dimensional structure of the full-length model of the Spike protein in open conformation derived by extensive massive all-atom molecular dynamics (MD) simulations of the glycosylated full-length model of the SARS-CoV-2 Spike protein embedded in a realistic compartment membrane/aqueous environment encompassing ∼1.7 million atoms [71]. The structural model (based on the PDB ID: 6VSB) [70] was downloaded from the Amaro lab (https://amarolab.ucsd.edu/covid19.php (accessed on 13 June 2021)), where computer simulations were developed with a near-atomic-scale resolution structure of viral components. These models are important for exploring the structure and dynamics of the virus and its interactions with the host cell, and also for the development of therapeutic options such as vaccines and antiviral drugs. This procedure opens up the possibility of performing mesoscale all-atom MD simulations of the complete virion. The protein is shown as ribbons, highlighting individual protein domains using different colors. The erected RBD in the open conformation state is also indicated. The Amaro lab has performed massive biophysical MD simulations, providing novel deep mechanistic insights into the molecular determinants playing a pivotal role in the virulence of the SARS-CoV-2 coronavirus, and especially to the SARS-CoV-2 Spike protein and its glycan coating, making a leap in strategizing a model for vaccine development, which is helpful in fighting the current global pandemic by providing more realistic data [73]. Further molecular simulation efforts focused on the intricacies of the binding of the SARS-CoV-2 RBD to ACE2 revealed critical hydrophobic regions and hydrogen-bonding networks [74].

The binding energies for the best docking pose of OEU and OC on the wild-type (wt) full-length Spike glycoprotein trimer of SARS-CoV-2 at the open conformation state (one RBD-up) (based on PDB ID: 6VSB) are summarized in Table 1. From Table 1, it is deduced that OEU exhibited better binding capacity compared to OC. The binding of OEU and OC on the crystal structure of the wt full-length model of the Spike protein in the open state is depicted in Figure 2. The selected OC/protein binding structure assembly is illustrated in Appendix A.

OC is stabilized at the interface between the NTD (14–305) (part of the S1) AS1 of protomer a (deep purple color) and the RBD domain (CS1) of protomer c (orange color). OC interactions involve binding with T114, Q115, N165, T167, G232, I233, and N234, of protomer a, and with N354, R355, K356, R357, and R466, of protomer c. OEU is stabilized away from furin cleavage site S1/S2 at the interface between the C-terminal domain 1 (CT1) of protomer b and the heptad repeat 1 (HR1) (912–984) and central helix (CH) domains of protomer c, being at the apical position of both the fusion peptide (FP) (816–855) and fusion peptide region (FPR) (856–911) of protomer c, flanked by the central helix (CH) (985–1034). OEU contacts involve G548, T549, A570, T572, T573, I587, and P589 residues of protomer b (deep teal color), and also M740, Y741, G744, D745, F855, N856, V963, L966, S967, and N978 of protomer c (orange color). Both OC and OEU interactions are reported in Appendix A.

#### 2.1.2. Wt Full-length Spike Closed (Based on 6VXX)

The binding energies for the best docking pose of OEU and OC on the wild-type (wt) full-length Spike glycoprotein trimer of SARS-CoV-2 at the closed conformation state (all RBDs-down) (based on PDB ID: 6VXX), are summarized in Table 1. From Table 1 it is deduced that OEU exhibited better binding capacity compared to OC. The binding of OEU and OC on the crystal structure of the wt full-length model of the Spike protein in the closed state is depicted in Appendix A. The selected OEU/protein binding structure assembly is illustrated in Appendix A.

OEU is located adjacent to S1/S2 furin cleavage site to S2′ (686–815) and in contact with the native D614 residue. It is also stabilized between the CT2 (591–685) of protomer B and the S2S2′ (686–815) and HR1 (912–984) domains and the upper part of FP (816–855) of protomer C. This native type of D614 residue makes the S protein less stable as compared to the D614G point mutation, which stabilizes it by blocking the premature shape change. Interestingly, the mutation also makes the S protein bind more weakly to the ACE2 receptor, but the fact that the spikes are less apt to fall apart prematurely (retaining their functionality), renders the virus overall more infectious. It is possible for OEU to interfere with the destabilization of the S protein by connecting to D614.

On the other hand, OC is placed in a binding site in proximity (binding R1000) to the central helix (CH) (985–1034) of the S2 subunit (flanked by its upper part) and also flanked by the upper part (binding L966, V976, L977) of HR1 (912–984) and the central part (binding with I746) of S2S2′ (686–815), and the upper part (binding with Asn N856) of FPR (856–911) of protomer A, as well as (binding with A570, T572) the CT1 (528–590) of protomer C adjacent to the polybasic (furin) cleavage site at the S1/S2 boundary. In Appendix A, it is shown that OEU interacts with the wt D614, stabilizing this structure. Since D614 is a mostly mutated site in the protein, also playing a role in the S1/S2 furin cleavage site—a most important event in the invasion process of the virus in human cells—it is interesting to compare the binding energies of OEU in both the wt and mutated variant D614G. The raised question is if there is an interaction of OEU on the fusion process of Spike protein (inhibition of fusion peptide). It is useful to examine if the mutated protein is more vulnerable to an OEU attack compared to the wt variant, resulting in a more stable (lower energy) binding complex.

### 2.2. Docking Calculations on Wild-Type and Mutant SARS-CoV-2 Spike and Its RBD Domain Proteins

SARS-CoV-2, as all viruses, changes over time. Most changes have little to no impact on the virus’ properties. The potential of the utilization of the two natural studied compounds, for the presumptive blocking of the Spike protein-dependent entry of SARS-CoV-2 into the host cell, is explored by performing molecular docking studies on both the wt and mt Spike proteins and their RBD domain. The ultimate goal is to validate the usefulness of the compounds in the designing of drug candidates for Spike target proteins and propose EVOO constituents OEU and OC as therapeutic targets for anti-COVID-19 drug development.

#### 2.2.1. Wt Open Spike Protein (One RBD-Up) (6VYB), (Two RBDs-Up) (7A93), Wt Closed Spike Protein (Three RBDs-Down) (6VXX)

The binding energies for the best docking pose of OEU and OC on the wild-type (wt) SARS-CoV-2 Spike glycoprotein trimer at both open conformation state with one and two RBDs-up (PDB IDs: 6VYB and 7A93, respectively) and closed state (all RBDs-down) (PDB ID: 6VXX) are summarized in Table 1. From Table 1, it is deduced that OEU exhibited better binding capacity compared to OC. The binding of OEU and OC molecules on the crystal structure of wild-type (wt) SARS-CoV-2 Spike protein in both closed (all RBDs-down, PDB entry code: 6VXX) and open (one and two RBDs-up, PDB entry codes: 6VYB and 7A93, respectively) conformation states is depicted in Figure 3a.

Comparing the binding architecture between the two open-conformation-state S proteins 6VYB and 7A93, it is deduced that OEU is placed in the RBD-up domain only in the 6VYB S protein. The schematic 2D and 3D binding interaction diagrams in Figure 3b are illustrating the binding interactions of OEU and OC with the 6VYB S protein’s residues of the binding site, generated with the aid of BIOVIA Discovery Studio 2016. The diagrams portray the interaction patterns between the ligands and the main-chain or side-chain elements of the protein. It should be noted that OC (in all 2D interaction diagrams) may appear bizarre due to the peculiar conformation shown, derived by bond rotations showing atoms and bonds to overlap. This is only a static representation and not the actual conformation of the molecule depicted in the 3D interaction diagram.

**Figure 3 molecules-27-07572-f003:**
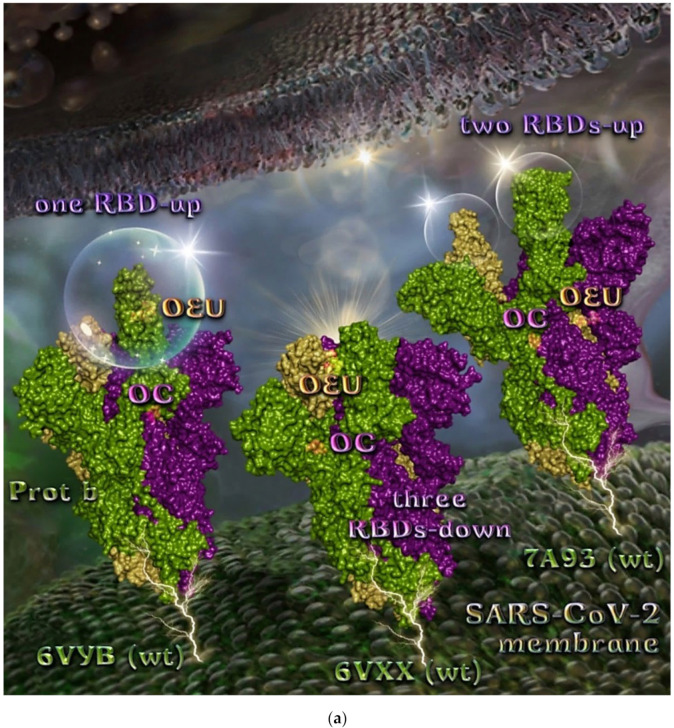
(**a**) Docking pose orientation of best-bound OEU and OC molecules on the crystal structure of wild-type (wt) SARS-CoV-2 Spike protein in both closed (all RBDs-down, PDB entry code: 6VXX) and open (one and two RBDs-up, PDB entry codes: 6VYB and 7A93, respectively) conformation states. Target trimeric wt S proteins are depicted in semitransparent surface representation colored by chain in deep purple, deep olive, and split pea green for each of the 3 protomers. The one and two RBDs-up domains are also indicated. OEU and OC are rendered in sphere mode and colored according to atom type in hot-pink and yellow-orange C atoms, respectively. N-linked glycans (NAG moieties) are omitted from the structure for clarity. Molecular docking simulations were performed individually. Hydrogen atoms are omitted from both molecules, and sugar molecules glycosylating the protein are hidden for clarity. Heteroatom color code: O—red. The final structure was ray-traced and illustrated with the aid of PyMol Molecular Graphics Systems. (**b**) Schematic 2D and 3D interaction diagrams showing the binding contacts of OEU and OC on 6VYB S protein. Residues rendered in either sphere or stick model are colored by interaction type and slate blue for 2D and 3D diagrams, respectively. OEU and OC rendered in line and ball-and-stick model (2D and 3D, respectively) are colored by atom type in grey and brown C atoms, respectively. Interactions are depicted in dotted lines colored according to interaction type. Solvent-accessible surfaces for each residue in 2D diagrams are depicted in light-blue spheres surrounding the residue spheres. The final structure was illustrated with the aid of BIOVIA Discovery Studio 2016.

#### 2.2.2. D614G mt Open Spike Protein (One RBD-Up) (7KDL)

During the early stages of the pandemic, the dominant variant, referred to as the D614G, was associated with high pathogenicity but without significant severity from its ancestral strain [76]. Binding energies for the best docking pose of OEU and OC on the mutant (mt) variant of the SARS-CoV-2 S protein trimer at open conformation state with one RBD-up bearing the D614G mutation, in which aspartic acid (D) residue in the 614 position is substituted by a glycine (G) residue (PDB entry code 7KDL), are summarized in Table 1. From Table 1, it is deduced that OEU exhibited better binding capacity compared to OC. The binding of OEU and OC molecules on the crystal structure of mutant (mt) variant of the SARS-CoV-2 S protein trimer bearing the D614G mutation (PDB entry code 7KDL) is depicted in Figure 4a. The diagrams in Figure 4b portray the schematic 2D and 3D binding interaction patterns of OEU and OC with the 7KDL S protein’s residues of the binding site, generated with the aid of BIOVIA Discovery Studio 2016.

The anchorage of OEU and OC is facilitated by the formation of hydrogen bond (H-bond), hydrophobic (Hph, alkyl-alkyl type), polar, π-polar, π-sulfur, and mixed π-type hydrophobic contact (π-alkyl type) interactions.

Additional binding interactions of OEU and OC not being portrayed by the 2D interaction diagrams but identified by PyMol, along with their binding details, are reported in Appendix A (D614G mt open Spike protein (one RBD-up) (7KDL)) of the Appendix A. Both OEU and OC seem to be stabilized inside the RBD in the “up” position exactly at the same place with the involvement of H-bond, π-polar, polar, Hph, and π-alkyl contacts, sharing a number of contacts, including **F338**, F342, **Y365**, L368, F377, K378, **P384**, **F392**, A397, **T430**, **C432**, V433, I434, V511, **L513**, and **F515** for OEU, and **F338**, L387, **Y365**, **P384**, **F392**, **T430**, **C432**, **L513**, and **F515** for OC (residues in boldface are common between OEU and OC).

Common binding contacts between OEU and OC are found to be P384, F392, T430, C432, L513, and F515. At the D614G binding site of OEU and OC, it is not unusual for conformational changes in this region to alter the conformation status of the RBD-up domain, causing possible alteration of its binding status to ACE receptor. This is indicative that the D614G substitution is directly associated with far-reaching alterations in interprotomer interactions and indirectly associated with changes in binding site exposure, leading to increased population of the one-up Spike ensemble [77]. It is well known that the D614G mutation triggers the change of RBD from its closed to a more open conformation that would make its binding to ACE2 more efficient [78].

#### 2.2.3. Mt Open Spike Protein (One RBD-Up) Alpha (8DLI), Beta (8DLL), Gamma (8DLO), Delta (7V7O), Epsilon (8DLT), and Kappa (7V7E)

Several variants of concerns (VOCs) have evolved from the original wild-type strain, such as lineages B.1.1.7 (Alpha variant with 17 mutations initially detected in the United Kingdom), B.1.351 (Beta variant with 9 mutations as a result of the second wave of COVID-19 infections in South Africa), and B.1.1.28.1 (Gamma variant with 10 mutations originating from Brazil). All of these variants harbor mutations in the N-terminal and receptor-binding domains of the Spike protein in which N501Y in the RBD is a common mutation to all variants [79]. The Delta variant, also known as the B.1.617.2, is first detected in India during the devastating wave of viral infection in April–May 2021.

Binding energies for the best docking pose of OEU and OC on the mutant (mt) open Spike protein variants (one RBD-up) Alpha (PDB ID: 8DLI), Beta (PDB ID: 8DLL), Gamma (PDB ID: 8DLO), Delta (PDB ID: 7V7O), Epsilon (PDB ID: 8DLT), and Kappa (PDB ID: 7V7E) of SARS-CoV-2 Spike trimeric glycoprotein are summarized in Table 1. From Table 1, it is deduced that OEU exhibited better binding capacity compared to OC for all variants.

The binding poses of OEU and OC molecules on the crystal structure of mutant (mt) variant of SARS-CoV-2 S trimeric proteins Alpha (PDB ID: 8DLI), Beta (PDB ID: 8DLL), Gamma (PDB ID: 8DLO), Delta (PDB ID: 7V7O), Epsilon (PDB ID: 8DLT), and Kappa (PDB ID: 7V7E) are depicted in Figure 5a. The docking procedure revealed OEU to be anchored in the RBD-up domain in Beta and Gamma S mt proteins, while in Alpha its stabilization is observed in the base of the RBD-up domain. OC and OEU were shown to be anchored at the interface between RBD and NTD, in a binding site of the S protein of Epsilon and Kappa variants in proximity to the central helix (CH) of the S2 subunit. The docking procedure demonstrated that both docked molecules are placed in a binding site in proximity to the central helix (CH) of the S2 subunit, at the base of the RBD domain and also adjacent to C-terminal domain 1 (CT1) of protomer a and the heptad repeat 1 (HR1) domain of protomer b. The diagrams in Figure 5b show the schematic 2D binding interactions of OEU and OC with Delta (7V7O) S protein’s residues of the binding site, generated with the aid of BIOVIA Discovery Studio 2016.

#### 2.2.4. Mt Open Spike Protein (One RBD-Up) Omicron BA.1 (7TGW, 7QO7), Omicron BA.2 (7XIW), Mt Closed Spike Protein (All RBDs-Down) Omicron BA.2.13 (7XNR), Omicron BA.3 (7XIY), and Omicron BA.4 (7XNQ, 7XNS)

The binding energies for the best docking pose of OEU and OC on the mutant (mt) open Spike protein variants (one RBD-up) Omicron BA.1 (PDB IDs: 7TGW, 7QO7) and BA.2 (PDB ID: 7XIW) as well as the mt closed Spike protein variants (all RBDs-down) Omicron BA.2.13 (PDB ID: 7XNR), BA.3 (PDB ID: 7XIY), and BA.4 (PDB IDs: 7XNQ, 7XNS) of SARS-CoV-2 Spike trimeric glycoprotein are summarized in Table 1. From Table 1, it is deduced that OEU exhibited better binding capacity compared to OC for all variants. The binding poses of OEU and OC molecules on the crystal structure of the mutant (mt) variants of SARS-CoV-2 S trimeric proteins are depicted in Figure 6a and Figure 7.

The docking procedure revealed OEU to be anchored in the RBD-up domain in Omicron BA.1 S mt protein. Both OEU and OC are stabilized in a binding cavity of the mt Spike Omicron BA.2 in open conformation state (one RBD-up) (PDB ID: 7XIW) (Figure 6a) in the vicinity of mutated residues G614, K764, H954, K969, and Y655 (at the furin cleavage site) [80]. Mutations of various residues often need to be introduced to improve protein expression or to trap the molecule in a particular state. Between all four Omicron subvariants—BA.1.1, BA.1, BA.2, and BA.3—common mutations identified were, among others, D614G, H655Y, N764K, N969K, G142D, G339D, S373P, S375F, K417N, N440K, S477N, T478K, E484A, Q493R, Q498R, N501Y, Y505H, N679K, P681H, D796Y, and Q954H.

Among G547, L764, and T856 contact residues of OEU and OC on Omicron BA.1 (7tgw), G547 and T856 were identified as mutated residues common among other sublineages. Other contacts of OEU in 7tgw were revealed to be T546, G547, I584, T570, P586, Y738, I739, C740, D742, M737, G741, F852, K853, R997, and L963, while OC was stabilized in its binding cavity by interacting with I309, S593, N610, A644, P662, I663, K730, L764, T765, T856, and L858. Two-dimensional schematic interaction diagrams showing the binding contacts of OEU and OC with Omicron BA.1 (7QO7) S protein’s residues of the binding site are depicted in Figure 6b.

**Figure 7 molecules-27-07572-f007:**
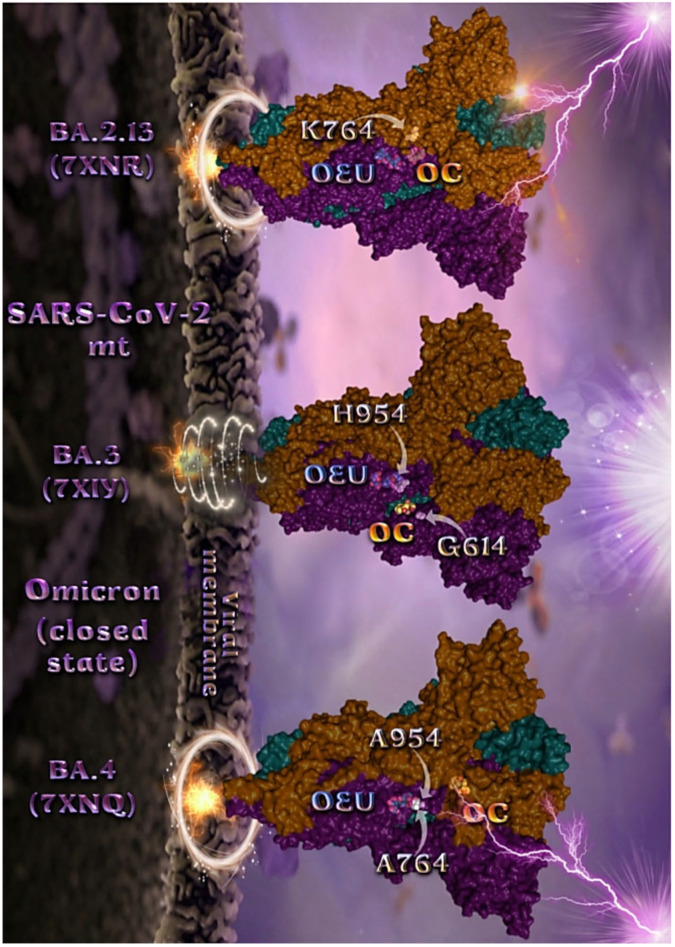
Docking pose orientation of best-bound OEU and OC molecules on the crystal structure of mutant (mt) Omicron BA.2.13 (7XNR), BA.3 (7XIY), and BA.4 (7XNQ, 7XNS) variants of SARS-CoV-2 S trimeric proteins in closed conformation state (all RBDs-down). The trimeric target proteins are illustrated as semitransparent surfaces with subdomains color-coded according to chain (protomers a, b, and c or chains A, B, and C in deep teal, deep purple, and orange, respectively). OEU and OC molecules (docked independently) rendered in sphere mode and colored according to atom type in cyan and yellow-orange C atoms, respectively. Mutated residues in the vicinity of the compounds’ binding pocket are rendered in white spheres in the mt proteins’ surfaces. In order to avoid cluttering of the structure, additional binding contact residues (reported in the Results section) are not displayed. Hydrogen atoms are omitted from both molecules, and sugar molecules glycosylating the protein are hidden for clarity. Heteroatom color code: O—red. The final structure was ray-traced and illustrated with the aid of PyMol Molecular Graphics Systems.

Mutated residues found in proximity to OEU and OC molecules on the crystal structure of the mutant (mt) Omicron variants of SARS-CoV-2 S trimeric proteins in open conformation state (one RBD in up position) (Figure 6) were identified to be G547, L764, and T856 for BA.1 (PDB ID: 7TGW); Q954, T856, F371, Y505, L764, T373, and K375 for BA.1 (PDB ID: 7QO7); and K764, K969, H954, G614, and Y655 (at the furin cleavage site) for BA.2 (PDB ID: 7XIW).

Mutated residues found in proximity to OEU and OC molecules on the crystal structure of the mutant (mt) Omicron variants of SARS-CoV-2 S trimeric proteins in closed conformation state (all RBDs-down) (Figure 7), were identified to be H954 and G614 for BA.3 (7XIY); K764 for BA.2.13 (7XNR); and A954 and A764 for BA.4 (7XNQ).

#### 2.2.5. Omicron BA.3 mt RBD of S Protein (7XIZ)

The binding energies for the best docking pose of OEU and OC on the mutant (mt) Omicron BA.3 variant of SARS-CoV-2 Spike trimeric glycoprotein’s RBD (PDB ID: 7XIZ) are summarized in Table 1.

From Table 1, it is deduced that OEU exhibited better binding capacity compared to OC. The binding of OEU and OC molecules on the crystal structure of the Omicron BA.3 variant of the Spike’s RBD is illustrated in Figure 8a.

The interacting residues of the Omicron BA.3 variant of the SARS-CoV-2 Spike trimeric glycoprotein’s RBD (7XIZ) with OEU and OC are presented in the 2D interaction diagrams of Figure 8b.

Both OEU and OC are shown to be stabilized at the interface between the RBD core and the RBM motif with the involvement of H-bond, π-polar, polar, Hph, π-anion, and π-alkyl contacts. OEU is predicted to be anchored in the vicinity of the base of a twisted five-stranded antiparallel sheet bundle (β1, β2, β3, β4, and β7) interacting with F342, N343, A344, T345, F371, P373, W436, N437, N439, K440, L441, and R509. Binding contacts of OC include the residues N394, R355, N394, P426, D428, F429, T430, F464, E516, and F515. Binding interaction details of OEU and OC are presented in Appendix A (Omicron BA.3 mt RBD of S protein (7XIZ)) of the Appendix A. All four Omicron lineages having common mutation at the receptor-binding motif (RBM) region (437–508a.a), which binds to hACE2, are N440K, S477N, T478K, E484A, Q493R, Q498R, N501Y, and Y505H [81]. Interestingly, OEU is shown to interact with K440 and thus interfere in the binding of RBM with hACE2. Furthermore, binding of OEU with Arg (N439) may destabilize the interaction between the S protein of SARS-CoV-2 and hACE2 by interfering in the hydrogen bonding between N439 of RBD and Glu (E329) of hACE2 [7].

### 2.3. Docking Calculations on Spike-ACE2, RBD-ACE2, and Spike-Monoclonal Antibodies, RBD-Monoclonal Antibodies Target Protein Complexes

#### 2.3.1. Spike-ACE2

The inhibition of spike-ACE2 protein–protein interaction using small molecules or peptides is the most logical and straightforward strategy to block viral cellular entry. Among a large number of potential targets, the inhibition of the direct interaction between ACE2 and the S protein provides a suitable strategy to prevent the membrane fusion of SARS-CoV-2 and the viral entry into human cells [82]. Spike is the main structural protein of coronavirus and assembles into a special corona structure on the surface of the virus as a trimer. Spike is a main protein that interacts with the host by binding to host cell receptors to mediate virus invasion. Small molecules, which can affect the binding efficiency of the Spike protein with its receptor, may act as the viral attachment inhibitor for SARS-CoV-2 [83,84]. Potential disruptors of the S glycoprotein interaction with ACE2, identified by simple molecular docking of approved drugs, include hesperidin [85], paritaprevir [86], cladribine, clofarabine, and fludarabine [87]. Docking studies with the complex Spike-ACE2 are employed in order to examine whether OEU and OC are able to disrupt the interaction of the Spike protein with ACE2 host receptor protein.

##### Wt Open Spike Protein/ACE2 Complex (7KJ2)

The binding energies for the best docking pose of OEU and OC on the protein complex between ACE2 and the wild-type (wt) of the Spike protein with one RBD in the up position (open conformation state) are summarized in Table 2. From Table 2, it is deduced that OEU exhibited better binding capacity compared to OC. The binding of OEU and OC with the native wt SARS-CoV-2 S trimer protein in complex with the ACE2 host (human) receptor protein (PDB ascension number 7KJ2) is shown in Figure 9a.

It is shown the open conformation state of the Spike protein with one RBD in up position bound to the ACE2 protein and the other two RBDs in down position. The protein is in its native state with no mutations (bears D614 and N501 wt residues). OEU (best lowest energy-binding pose) and OC in its higher energy-binding pose are shown to be anchored at the interface between RBD and NTD and in more proximity to the latter.

On the other hand, OC in its lowest energy-binding pose is placed in a pocket inside the connecting domain (CD) in contact with D1118 residue. OC and OEU are placed in a binding site in proximity to the central helix (CH) of the S2 subunit. The docking procedure demonstrates that both docked molecules are placed in a binding site in proximity to the central helix (CH) of the S2 subunit, at the base of the RBD domain and also adjacent to C-terminal domain 1 (CT1) of protomer a and the heptad repeat 1 (HR1) domain of protomer b, while OC in its lowest energy-binding pose is stabilized between the connector domain (CD1) (1081–1147) and HR1 (912–984) of protomer a and CD1 (1081–1147) of protomer c. OC is also making contact with the mutated residue D1118 of protomer a. OC is also predicted to interact with C749 of protomer a belonging to the S2S2′ (686–815) domain. Interestingly, OC in its lowest energy-binding pose is making contact with the mutated residue D1118 of protomer a. Vaccine targeting segments 884–891 and 1116–1123 in S2 domain were highly effective in inducing humoral and cell-mediated immune responses [88]. These segments belong to the central helix between HR1 and HR2. The HR1 and HR2 domains, involved in viral fusion, and the CD domain, which connects the two heptad repeats [70] helping in the stabilization of postfusion structure [89], constitute important regions of S protein as hot spots for target therapy.

The interacting residues of wild-type (wt) SARS-CoV-2 Spike protein in complex with the ACE2 host (human) receptor protein (7KJ2) with OEU and OC are depicted in the 2D interaction diagrams of Figure 9b.

The docking procedure predicts the formation of a variety of interactions such as H-bond, π-polar, polar, Hph, and π-alkyl contacts between OEU and the amino acid residues A520, H519, R567, D571, T573, L517, C391, T393, N544, G545, L546, and T547 of protomer a, and V976, N978, D979, and S982 of protomer b. OC and OEU are sharing common binding contacts since they are anchored exactly at the same place. The binding interaction details of OEU and OC are presented in Appendix A (Wt open Spike protein/ACE2 complex (7KJ2)) of the Appendix A.

**Figure 9 molecules-27-07572-f009:**
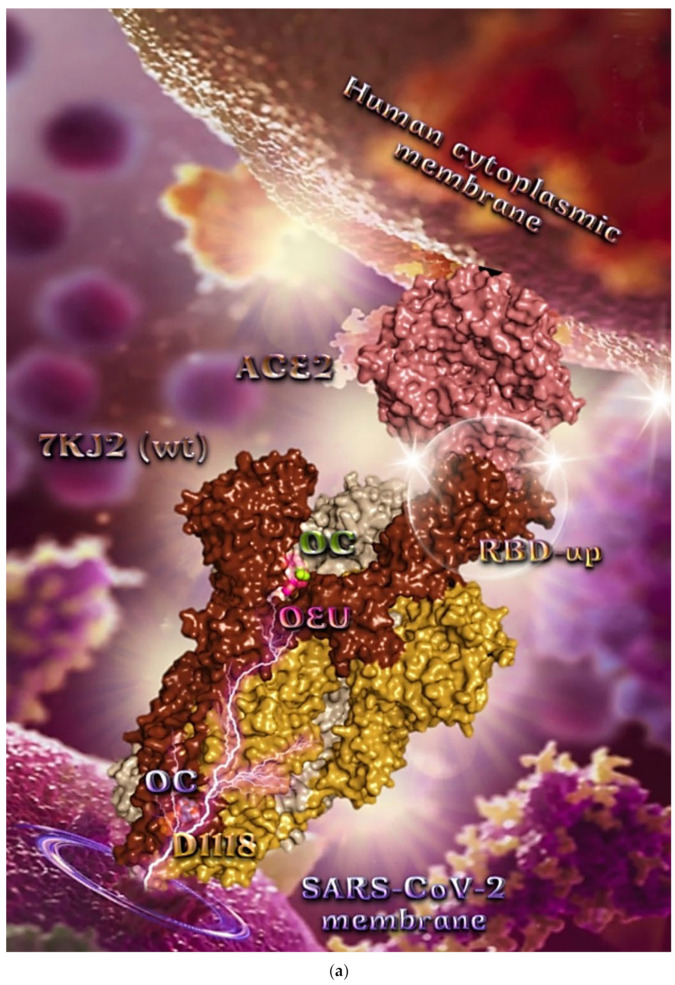
(**a**) Docking pose architecture of best-bound OEU and OC molecules, on the crystal structure of wild-type (wt) SARS-CoV-2 Spike protein in complex with the ACE2 host (human) receptor protein (PDB entry code 7KJ2). The trimeric target protein is illustrated as semitransparent surface color-coded according to chain (protomers a, b, and c in chocolate, bright orange, and wheat color, respectively, and chain D of ACE2 in salmon). The protein is illustrated in its open conformation state with one RBD in up position. The one RBD-up domain is also indicated inside a transparent sphere. Both docked molecules are rendered in sphere mode colored by atom type in hot pink (OEU) and split-pea green and slate blue (OC on higher and lowest energy-binding poses, respectively). OEU and OC (higher energy-binding pose) are placed in a binding site in proximity to the central helix (CH) of the S2 subunit, at the base of the RBD domain and also adjacent to C-terminal domain 1 (CT1) of protomer a and the heptad repeat 1 (HR1) domain of protomer b, while OC in its lowest energy-binding pose is stabilized between the connecting domain (CD1) (1081–1147) and HR1 (912–984) of protomer a and CD1 (1081–1147) of protomer c. OC is also making contact with the mutated residue D1118 of protomer a. OC also makes contact with C749 of protomer a belonging to the S2S2′ (686–815) domain. Hydrogen atoms are omitted from all molecules, and sugar molecules glycosylating the protein are hidden for clarity. Heteroatom color code: O—red. The final structure was ray-traced and illustrated with the aid of PyMol Molecular Graphics Systems. (**b**) Schematic 2D interaction diagrams showing the binding contacts of OEU and OC on wild-type (wt) SARS-CoV-2 Spike protein in complex with the ACE2 host (human) receptor protein (PDB entry code 7KJ2). Solvent-accessible surfaces for each residue are depicted in light-blue spheres. The final structure was illustrated with the aid of BIOVIA Discovery Studio 2016.

##### Omicron BA.2 mt Spike Protein/ACE2 Complex (Two and Three ACE2 Bound) (7XO7 and 7XO8)

The binding energies for the best docking pose of OEU and OC on the protein complex between ACE2 and the mutant (mt) variants of Omicron BA.2 Spike protein with two and three ACE2 proteins bound (7XO7 and 7XO8, respectively) are summarized in Table 2. From Table 2, it is deduced that OEU exhibited better binding capacity compared to OC. The binding of OEU and OC molecules on the crystal structure of BA.2 mt Spike protein/ACE2 complexes is illustrated in Figure 10a.

OEU is stabilized at the interface between the central helix (CH) and the HR1 domains of protomer a and C-terminal domain 1 (CT1) of protomer c of 7xo7. OEU and OC interactions include π-π displaced, H-bond, π-polar, polar, Hph, and π-alkyl contacts. OEU is stabilized via binding with M731, K947, D950, H954, Q957, Q1010, R1014, E1017, and I1018 of protomer a (purple blue); and Q762, R765, A766, G769, E773, V951, R1019, and I312 of protomer c (purple) for OEU. On the other hand, OC is stabilized at the interface between the S1/S2 furin cleavage site, the fusion peptide (FP), and the S2′ domains. OC is positioned in its binding pocket with the incorporation of I312, Q314, S596, Q613, P665, and I666 of protomer a (purple-blue); and additionally T768, V772, and D775 of protomer a (purple-blue). The binding interaction details of OEU and OC are presented in Appendix A (Omicron BA.2 mt Spike protein/ACE2 complex (two and three ACE2-bound) (7XO7 and 7XO8)) of the Appendix A. Two-dimensional schematic interaction models showing the binding contacts of OEU and OC with the Omicron BA.2 variant of SARS-CoV-2 Spike trimeric glycoprotein in complex with two human ACE2 receptor proteins (PDB entry code 7XO7) are depicted in Figure 10b.

**Figure 10 molecules-27-07572-f010:**
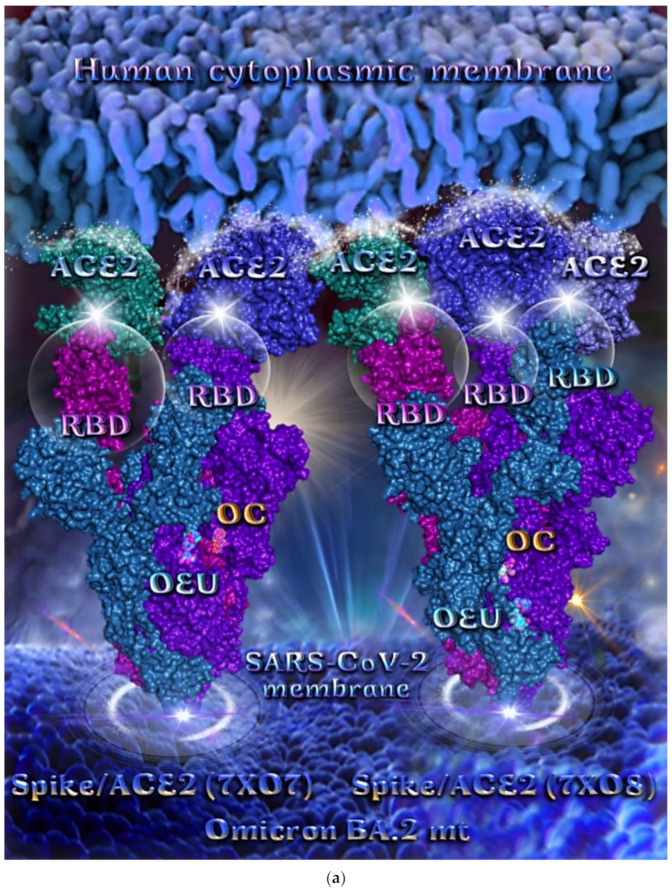
(**a**) Docking pose architecture of best-bound OEU and OC molecules on the crystal structure of mutant (mt) Omicron BA.2 variant of SARS-CoV-2 Spike trimeric glycoprotein in complex with two (PDB entry code 7XO7) or three (PDB entry code 7XO8) human ACE2 receptor proteins. The trimeric target proteins are illustrated in semitransparent surface model color-coded according to chain (protomers a, b, and c in purple-blue, sky-blue, and purple color, respectively, and chains E, F and E, F, D of ACE2 in teal, tv blue, and teal, tv blue, and slate blue, respectively). The proteins are illustrated in their open conformation state with two or three RBDs in up position indicated inside a transparent sphere. Both docked molecules are rendered in sphere mode colored by atom type in cyan (OEU) and yellow-orange (OC). Hydrogen atoms are omitted from all molecules, and sugar molecules glycosylating the protein are hidden for clarity. Heteroatom color code: O—red. The final structure was ray-traced and illustrated with the aid of PyMol Molecular Graphics Systems. (**b**) Schematic 2D interaction diagrams showing the binding contacts of OEU and OC on Omicron BA.2 variant of SARS-CoV-2 Spike trimeric glycoprotein in complex with two human ACE2 receptor proteins (PDB entry code 7XO7). Solvent-accessible surfaces for each residue are depicted in light-blue spheres. The final structure was illustrated with the aid of BIOVIA Discovery Studio 2016.

#### 2.3.2. RBD-ACE2

Molecular docking studies are adopted in both the wt and mt RBD/ACE2 complex to explore the ability of the studied EVOO constituents to interfere with this protein complex, and the designing of a future class of RBD/ACE2 blockers.

##### Wt Full-Length S Protein’s RBD/ACE2 Complex (from 6M17)

An ideal drug candidate should selectively target the RBD without interacting with ACE2 to avoid possible side effects linked to angiotensin physiology [90]. Docking studies with the complex RBD-ACE2 are employed in order to examine whether OEU and OC are able to disrupt the interaction of RBD with ACE2 host receptor protein.

Binding energies for the best docking pose of OEU and OC on the protein complex between ACE2 (acting as the receptor to infect human cells) and the wild-type (wt) of RBD are summarized in Table 2. From Table 2, it is deduced that OEU exhibited better binding capacity compared to OC, although they displayed similar binding architecture since they are stabilized at the same place of the interface between the RBD-ACE2 protein complex. Docking pose orientations of compounds OEU and OC are depicted in Figure 11. A close-up view of the ligand-binding architecture sites of OEU and OC in the RBD-ACE2 complex, depicting the extent of the binding pocket as determined by the computation process as well as the crystal structure, are depicted in Figure 12 and Figure 13, respectively.

The anchorage of OEU and OC is facilitated by the formation of H-bond, hydrophobic (alkyl-alkyl type), polar, π-polar, mixed π-type hydrophobic contacts (π-alkyl type), π-π displaced, and π-cation and π-anion electrostatic interactions. Amino acid residues participating in these interactions contributing to binding affinity between RBD and ACE2 include D30, H34, E35, E37, D38 (belonging to the ACE2 receptor); and R402, R403, K417, Y453, Q493, S494, Y495, and Y505 (belonging to the RBD domain).

**Figure 12 molecules-27-07572-f012:**
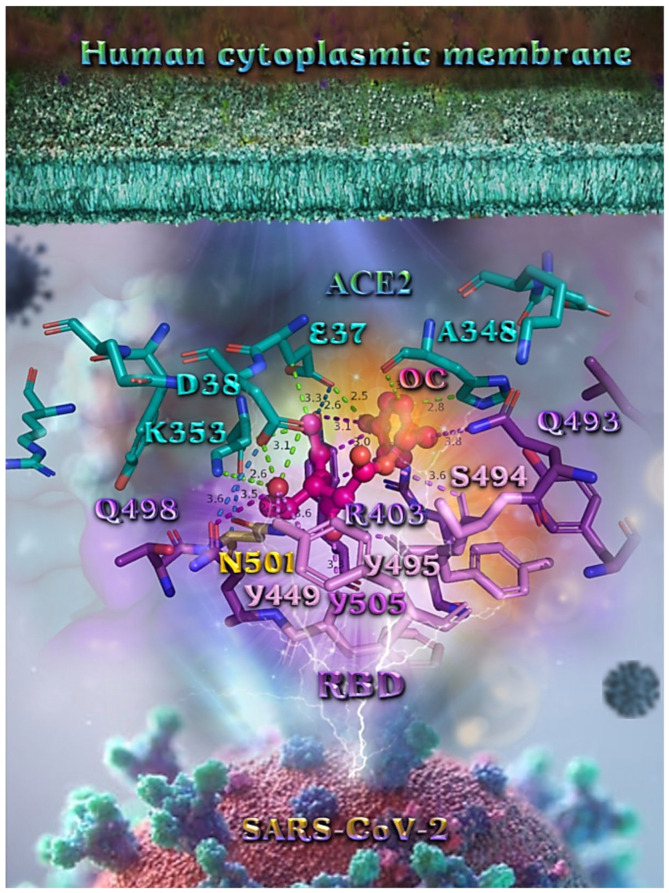
A close-up view of the binding site mapping interactions of OC at the interface between RBD domain of SARS-CoV-2 full-length S protein and the ACE2 receptor. ACE2 and RBD proteins are illustrated as semitransparent surfaces in deep purple for RBD and deep teal for ACE2. OC is rendered in ball-and-stick mode colored by atom type in hot-pink C atoms. Binding residues are illustrated in stick model colored by atom type according to the surfaces’ colors. N501 wt residue is depicted in bright-orange stick model. Binding interactions of OC are depicted in chartreuse-green dotted lines with ACE2 and in purple dotted lines with RBD, while contacts between RBM-ACE2 are indicated in marine-blue dotted lines. Hydrogen atoms are omitted from all molecules, and sugar molecules glycosylating the protein are hidden for clarity. Heteroatom color code: O—red. The final structure was ray-traced and illustrated with the aid of PyMol Molecular Graphics Systems.

Likewise, the OC interactions with ACE2 were predicted to involve H34, E37, D38, K353, (belonging to the ACE2 receptor); and R403, Y449, Q493, Y495, Q498, N501, and Y505 (belonging to the RBD domain). A significant role in the anchorage of OC is also played by the aromatic His (H34) residue mediating π-π (T-shaped, 2.8 Å) interactions with the phenol aromatic ring moiety of OC, contributing thus to an additional stabilization of the docked molecule in the protein.

**Figure 13 molecules-27-07572-f013:**
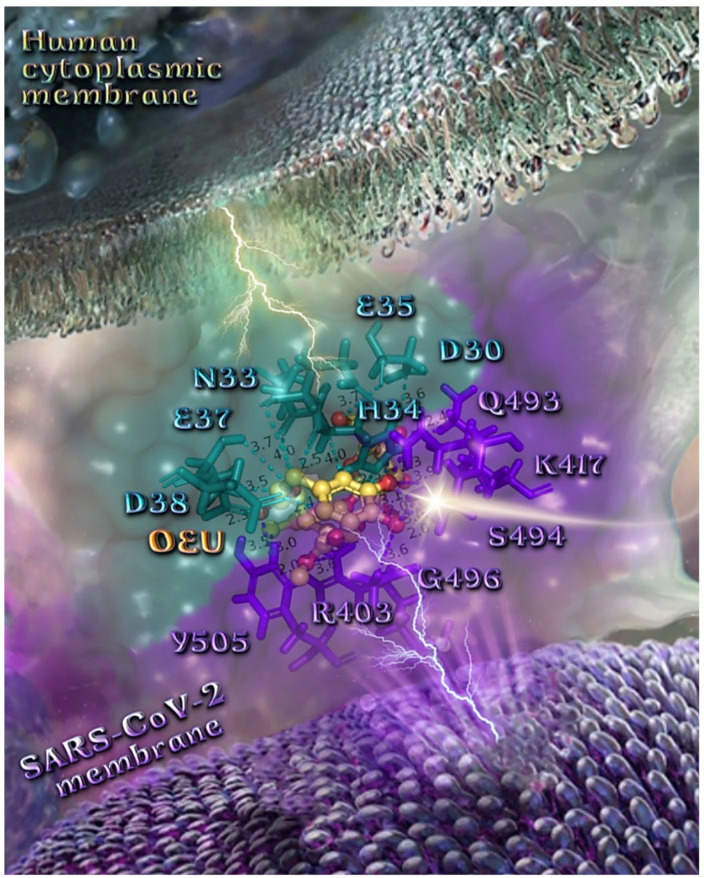
A close-up view of the binding site mapping interactions of OEU at the interface between the RBD domain of SARS-CoV-2 full-length S protein and the ACE2 receptor. ACE2 and RBD proteins are illustrated as semitransparent surfaces in deep purple for RBD and deep teal for ACE2. OEU is rendered in ball-and-stick representation colored by atom type in yellow-orange C atoms. Binding residues are illustrated in stick model colored by atom type according to the surfaces’ colors. Binding interactions of OEU are depicted in chartreuse-green dotted lines with ACE2 and in purple dotted lines with RBD, while contacts between RBM-ACE2 are indicated in marine-blue dotted lines. Hydrogen atoms are omitted from all molecules, and sugar molecules glycosylating the protein are hidden for clarity. Heteroatom color code: O—red. The final structure was ray-traced and illustrated with the aid of PyMol Molecular Graphics Systems.

Binding interaction details of OEU and OC are presented in Appendix A (Wt full-length S protein’s RBD/ACE2 complex (from 6M17)) of the Appendix A.

The specific residue pair interactions in the RBD/ACE2 interface, driving their protein–protein interaction, involve the contacts (RBD residue—ACE2 residue): L455-**H34**, Y489-Y83, T500-Y41, **N501**-Y41, **Y505**-**E37**, **Y505**-N393, **Y449**-**D38**, **Y449**-Q42, G496-**K353**, **Y449**-**D38**, **Y453**-**H34**, N487-Q24, N487-Y83, A475-S19 (starting of N-terminal helix), R439-E329, T446-Q42, G502-**K353** (belonging to 325-loop), Q496-K31, and Q496-**E35**. Notably, D30 is found to be one of the binding interactions of OEU with the interface of ACE2 via the hydrogen bond contact. Moreover, the D30 and K26 residues of ACE2 play a critical role in the interaction between RBD and ACE2. The D30, K31, H34, and D38 residues are part of the central segment of the α1 helix.

For this reason, these residues have the potential to be developed as a target for entry inhibitors [91]. Moreover, D38, being a common contact of both OEU and OC, is a key binding site that forms hydrogen bonds with Y449 of RBD [92]. Therefore, these Asp residues can be used as primary target active sites of ACE2 inhibitors. ACE2 can be a target for inhibiting the entry of SARS-CoV-2 into the host cell because the binding affinity of the S protein to the ACE2 receptor is 10–20-fold stronger than that of the S protein of SARS-CoV [93].

From the above, it is obvious that both OEU and OC seem to interfere in the interaction between the interface between RBD-ACE2 mediated by the residues indicated in boldface. It is interesting to notice that OEU interferes with RBD-ACE2 binding via four out of a total of five contacts with ACE2 indicated in boldface (D30, **H34**, **E35**, **E37**, and **D38**), while OC interferes with the incorporation of all of its four contacts (**H34**, **E37**, **D38**, and **K353**). On the other hand, the corresponding interfering binding contacts of OEU with the RBD domain number only two out of a total of eight contacts, indicated in boldface (R402, R403, K417, **Y453**, Q493, S494, Y495, and **Y505**), while OC mediates its interference via three out of a total of seven contacts, indicated in boldface (R403, **Y449**, Q493, Y495, Q498, N501, and **Y505**).

Interestingly, the H34, N33, K417, and Y505 binding contacts of OEU (Figure 13) are found to be in common with those observed for the oleuropein dimer on the interface RBD/ACE2 [64], while D30, E37, and N33 are also observed as binding contacts of the phytochemicals ursolic acid, maslinic acid, and glycyrrhizinic acid [15].

It should also be noted that the RBD residues’ Q493 and Q498 binding contacts of OEU and OC were identified as key locations for the SARS-CoV-2 host range [81]. Furthermore, the R403, Y505, N501, Y495, Q493, H34, E37, D38, and K417 binding contacts of both OEU and OC are also observed as binding interactions for the RBD-ACE2 complex (PDB 6M17) of the approved small molecules cefsulodin, cromoglycate, nafamostat, nilotinib, penfluridol, and radotinib [94].

##### Wt S Proteins’ RBD/ACE2 Complex (6VW1)

The binding energies for the best docking pose of OEU and OC on the protein complex between the wild-type (wt) of RBD and ACE2 (PDB ID: 6VW1) are summarized in Table 2. From Table 2, it is deduced that OEU exhibited better binding capacity compared to OC. The binding of OEU and OC on wt S proteins’ RBD/ACE2 complex is shown in Figure 14a.

OEU is stabilized in the same region of the protein where the MLN-4760 inhibitor finds accommodation, inside a binding cleft formed above α2 helix, and the β3, β4 beta strands of ACE2, adjacent to the ridge of the N-terminal α1 helix of ACE2 responsible for the binding of RBD to ACE2. OEU and OC make numerous contacts with the participation of the H-bond, π-polar, polar, Hph, π-anion, and π-alkyl contacts. OEU is stabilized via contacts with T276, D292, M366, D367, L370, T371, **E406**, **S409**, L410, **K441**, **Q442**, and **T445** residues. OC is anchored in the same binding pocket with OEU, with the inclusion of residues **E406**, **S409**, A413, F438, **Q442**, **K441**, **Q442**, **T445**, and I446. Common binding contacts between OEU and OC are indicated in boldface type. The binding interaction details of OEU and OC are presented in Appendix A (Wt S proteins’ RBD/ACE2 complex (6VW1)) of the Appendix A.

**Figure 14 molecules-27-07572-f014:**
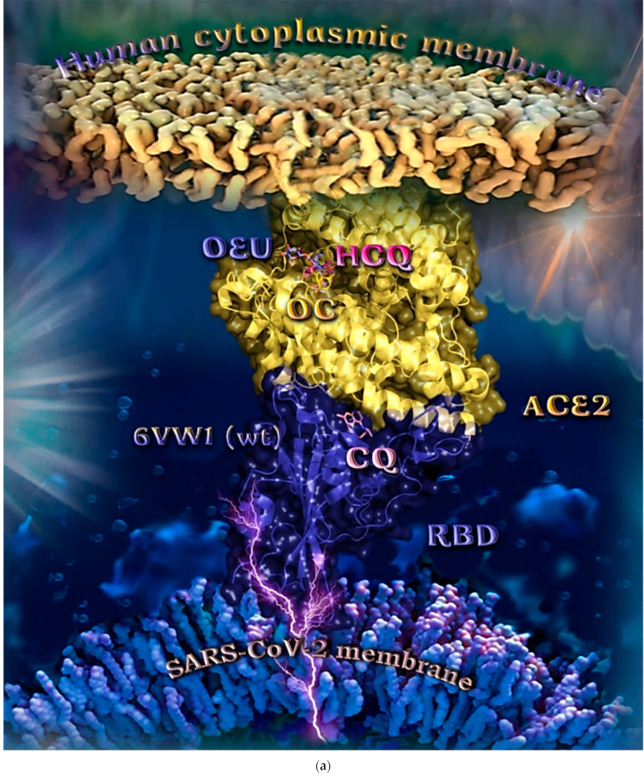
(**a**) Docking pose orientation of best-bound OEU and OC molecules on the crystal structure of SARS-CoV-2 S protein’s RBD bound to the ACE2 receptor (PDB: 6VW1 native wild-type (wt) protein structure). ACE2 and RBD proteins are illustrated as yellow and deep-blue cartoons, respectively. All docked molecules, OEU, OC, chloroquine (CQ), and hydroxychloroquine (HCQ) are rendered in stick representation colored by atom type in slate-blue, yellow, salmon, and hot-pink C atoms, respectively. CQ is shown to be anchored at the interface of RBD-ACE2 complex, while OEU and OC are shown to be stabilized in a binding cavity of the ACE2 receptor at the same place occupied by the ACE2 inhibitor HCQ. Molecular docking simulations were performed individually. Hydrogen atoms are omitted from both molecules for clarity. Heteroatom color code: O—red, N—blue. The final structure was ray-traced and illustrated with the aid of PyMol Molecular Graphics Systems. (**b**) Schematic 2D interaction diagrams showing the binding contacts of OEU, OC, CQ, and HCQ on SARS-CoV-2 S protein’s RBD bound to the ACE2 receptor (PDB: 6VW1 native wild-type (wt) protein structure). Solvent-accessible surfaces for each residue are depicted in light-blue spheres. The final structure was illustrated with the aid of BIOVIA Discovery Studio 2016.

Chloroquine (CQ) is shown to be anchored at the interface of RBD-ACE2 complex, while OEU, OC, and hydroxychloroquine (HCQ) are shown to be stabilized in a binding cavity of the ACE2 receptor adjacent to the place occupied by the ACE2 inhibitor MLN-4760 [95].

Docking results, therefore, highlight the potential role of OEU and OC as ACE2 inhibitors. However, the reduction in ACE2 activity is detrimental to the heart, since it contributes to cardiac dysfunction, partly due to the increased stimulation of the AT1 receptor by angiotensin II [96]. Two-dimensional schematic interaction models showing the binding contacts of OEU and OC with native wt SARS-CoV-2 S protein’s RBD bound to the ACE2 receptor (6VW1) are illustrated in Figure 14b.

##### Delta and Kappa S Proteins’ RBD/ACE2 Complex (7V8B and 7V87)

A series of in silico studies were employed in order to predict the biological activity of the studied compounds OEU and OC on Delta and Kappa mutant SARS-CoV-2 RBD-ACE2 conjugate structures (PDB IDs: 7V8B and 7V87, respectively). Docking studies were performed with a view to find a possible effect of these compounds on the binding interaction between RBD and ACE2 host receptor protein. Binding energies for the best docking pose of OEU and OC on the protein complex between ACE2 and the mutant (mt) Delta and Kappa S proteins’ RBD (PDB IDs: 7V8B and 7V87, respectively) are summarized in Table 2. From Table 2, it is deduced that OEU exhibited better binding capacity compared to OC for the Kappa mutant variant and the same for Delta. The best-fitted docking poses of OEU and OC, exhibiting the highest in silico binding capacity on RBD/ACE2 complexes of the mt Delta and Kappa variants, are shown in Figure 15a.

OEU is anchored at the interface between the RBD domain and ACE2 protein of the Delta variant adjacent to R452 mutant residue. The L452R mutation of the Delta mutant variant, bearing the R452 mutated residue compared to the L452 residue of wt, is located in the RBD domain of the S protein and may stabilize the interaction between the S protein and its human ACE2 receptor, and thereby increase infectivity [97]. Similar binding at the interface between RBD and ACE2 is also identified for the (S)-enantiomer of linezolid [98], which interacted with both the RBD domain of the 6VW1 wt Spike protein and ACE2 receptor via Tyr Y453 and His H34, respectively. Furthermore, the E37 binding contact of (S)-Linezolid with the ACE2 protein is also found in proximity to the binding site of both OEU and OC, making contact with them. OC is stabilized inside a cavity in ACE2 adjacent to the place occupied by the ACE2 inhibitor MLN-4760 positioned inside a pocket above the α2 helix, but not sharing common contacts with it. OEU is stabilized below the central part of α1 helix of ACE2 in an elongated crevice formed additionally by the β strands β5 and β6 of the RBM motif and helix α4 of RBD, at the apical position of the β3 strand of RBD-core, encircled by the ligand-binding site contact residues of both RBD and ACE2. OC is surrounded by helices α9 and α10, the edge of the α12, α13, and α14 helices, and the loop region between helices α8 and α9.

The binding interaction details of OEU and OC are presented in Appendix A (Delta and Kappa S proteins’ RBD/ACE2 complex (7V8B and 7V87)) of the Appendix A. OEU and OC make numerous contacts with the participation of H-bond, π-π displaced (offset), π-π sandwich, π-polar, polar, Hph, π-anion, and π-alkyl contacts. OEU is stabilized via contacts with Y505, Y495, R403, E406, K417, Y453, R454, L455, S494, S494, Y495, and Y449 of RBD, and H34, E37, D38, and K353 of ACE2.

The docking procedure reveals the binding of OEU with H34 residue of ACE2, which is normally in contact with Y453 of RBD, thus perturbing this stabilization interaction between the two proteins. The same is valid for the following pair interaction in the stabilization of the RBD/ACE2 protein complex: Y449/D38 (binding of OEU with both D38 and Y449), Y505/E37 (binding of OEU with both residues), Y449/K353 (binding of OEU with both residues), Y453/H34 (binding of OEU with both residues), and G496/K353 (binding of OEU with both residues). All the observed distortions of RBD/ACE2 pairings seem to disrupt the canonical stabilization of the protein complex.

The interacting residues of Omicron BA.3 variant of SARS-CoV-2 S protein’s RBD of Delta and Kappa variants of SARS-CoV-2 in complex with the ACE2 host receptor protein (7v8b and 7v87, respectively) with OEU and OC are presented in the 2D interaction diagrams of Figure 15b,c.

OC is anchored at the interface between the RBD domain and ACE2 protein of Kappa variant (7v87), while OEU is buried deep in the binding pocket of ACE2 accommodating MLN-4760. On the contrary, while OC and OEU are stabilized at almost the same place of ACE2-binding pocket of the Delta (7v8b) and Kappa (7v87) variants, respectively, OEU succeeded in entering deep in the binding cleft inserted in the crevice along its whole depth, compared to OC, which is shown to be anchored at the entrance of the pocket.

**Figure 15 molecules-27-07572-f015:**
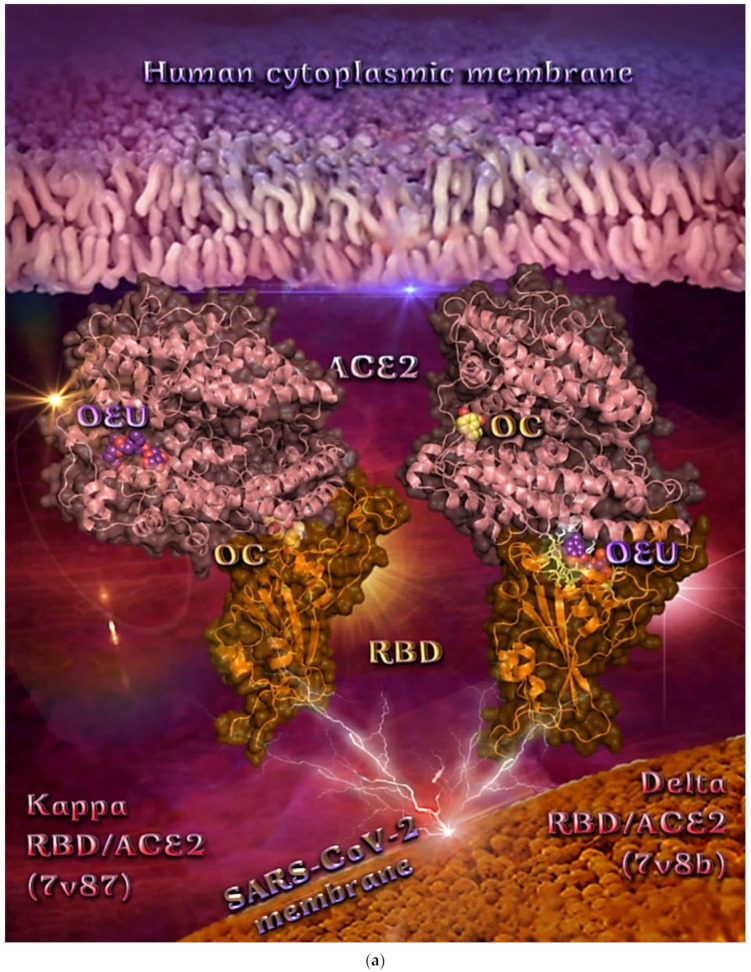
(**a**) Binding pose architecture of OEU and OC on the crystal structure of SARS-CoV-2 S protein’s RBD in complex with ACE2 host receptor protein (PDB IDs: 7v8b and 7v87 for Delta and Kappa mutation variants of SARS-CoV-2, respectively). ACE2 protein is illustrated as deep-salmon cartoons, while RBD protein is depicted in orange cartoon with additional depiction, for both proteins, of semitransparent surface colored according to cartoon colors. OEU and OC are rendered in sphere mode colored by atom type in violet-purple and yellow-orange, respectively. The ligand-binding site contact residues of OEU inside the interface of Delta RBD/ACE2 variant are indicated in yellow-orange (RBD) and white (ACE2) sticks, respectively. Hydrogen atoms are omitted from all molecules, and sugar molecules glycosylating the protein are hidden for clarity. Heteroatom color code: O—red. The final structure was ray-traced and illustrated with the aid of PyMol Molecular Graphics Systems. (**b**) Schematic 2D interaction diagrams showing the binding contacts of OEU and OC on the crystal structure of SARS-CoV-2 S protein’s RBD of Delta variant of SARS-CoV-2 in complex with ACE2 host receptor protein (PDB IDs: 7v8b). The final structure was illustrated with the aid of BIOVIA Discovery Studio 2016. (**c**) Schematic 2D interaction diagrams showing the binding contacts of OEU and OC on the crystal structure of SARS-CoV-2 S protein’s RBD of Kappa variant of SARS-CoV-2 in complex with ACE2 host receptor protein (PDB IDs: 7v87). The final structure was illustrated with the aid of BIOVIA Discovery Studio 2016.

OC at the interface between the RBD domain and ACE2 protein of the Kappa (7v87) variant seems to be encircled by the β strands β5 and β6 of the RBM motif and the α1 N-terminal helix of ACE2, making contact with the N33, H34, E37, N388, P389, and R393 residues of ACE2 and the E406, Y453, and Y453 residues of RBD.

In a similar binding mode with OEU (on the Delta variant), OC (on the Kappa variant) seems to intervene in the RBD/ACE2 pairing contact interacting with H34, E37, R393 of ACE2 and Y453 of RBD, destabilizing the pairings Y453/H34, Y505/E37, and Y505/R393. Two hydrophobic contacts of (S)-Linezolid with ACE2 were also found to be common with those of OC, namely Asn N33 and Pro P389. Furthermore, binding contacts N33, H34, D38, Y453, and P389 (with the latter playing a critical role in the interaction of both OEU and OC on ACE2), were revealed to be in common with luteolin [99,100]; N33, H34, and P389 were in common with andrographolide; H34, A387, and P389 were in common with artemisinin; and H34 was in common with pterostilbene [26], bound at the same RBD/ACE2 interface.

R393 and P389 binding residues of both OEU and OC are also reported as binding contacts of maslinic acid and epoxyazadiradione phytochemical compounds on the RBD/ACE2 interface [15]. Furthermore, R403, L455, Y453, Y495, G496, K417, K353, Y505, H34, and E37 binding contacts of OEU and OC have also been found to be common with the approved small molecules cefsulodin, cromoglycate, nafamostat, nilotinib, penfluridol, and radotinib [94]. Furthermore, binding contact residues of OEU and OC, documented also as common contacts with andrographolide (N33, P389, R393, Y505, and H34), pterostilbene (H34, S494, and G496), and resveratrol (G496) phytochemical compounds [26].

##### Omicron BA.1 and BA.2 mt S Proteins’ RBD/ACE2 Complex (PDB Ascension N’s 7WPB and 7XO9)

The SARS-CoV-2 Omicron variant (lineage B.1.1.529, South Africa/Botswana, including its sublineages BA.1 and BA.2) was reported to the World Health Organization (WHO) in late November 2021 in South Africa and has become the dominant infective strain, accounting for nearly all sequences reported to GISAID. This variant of SARS-CoV-2 accumulates an unprecedentedly high number of mutations, most of which are located on the surface of the Spike protein, and especially on RBD, compared to former variants involved in the COVID-19 pandemic, thus changing binding epitopes to many current antibodies strengthening the RBD binding to ACE2 [101].

Omicron evolved independently from previous VOCs, including the predominant Alpha, Beta, Gamma, and Delta variants [102]. Compared to the original wild-type (wt) strain of SARS-CoV-2, Omicron has 60 amino acid mutations, of which 37 mutations are in the Spike protein, the target of most COVID-19 vaccines and therapeutic antibodies [103]. This high variation is reflected in different behaviors, with the Omicron variant showing enhanced transmission, antibody evasion, and vaccine resistance [104,105]. The reported mutations are the following: A67V (BA.1 only), Δ69-70, T95I, G142D, Δ143-145, Δ211-212, ins214EPE, G339D, S371L, S373P, S375F, K417N, N440K, G446S, S477N, T478K, E484A, Q493K, G496S, Q498R, N501Y, Y505H, T547K, D614G, H655Y, N679K, P681H, N764K, D796Y, N856K, Q954H, N969K, L981F. The Omicron RBD forms extra interactions with ACE2, including interactions from RBD mutations S477N, Q493R, Q498R, and N501Y to ACE2.

The binding energies for the best docking pose of OEU and OC on the protein complex between ACE2 and the mutant (mt) Omicron BA.1 and BA.2 mt S proteins’ RBD (PDB IDs: 7WPB and 7XO9, respectively) are summarized in Table 2. From Table 2, it is deduced that OEU exhibited better binding capacity compared to OC for the BA.2 variant and the same for the BA.1 variant. The docking procedure demonstrated similar binding of both compounds on Omicron BA.1 and BA.2 mt S proteins’ RBD/ACE2 complex (PDB ascension N’s 7WPB and 7XO9, respectively).

Both OEU and OC are shown to be anchored at the same binding cavity in ACE2, sharing common binding contacts and exhibiting similar binding capacity (almost the same binding energy) (Figure 16a). They are placed above the α2 helix of ACE2 in close contact to the edges of α3, α6, and α7 helices and in proximity to the loop between α12 and α13, the edge of α19 and the loop between α18 and α19. Among other, the molecule binding of both OEU and OC with loops that are known to play a major role in the stability of the protein structure [106] could potentially affect the protein stability.

Two-dimensional schematic interaction models showing the binding contacts of OEU and OC with the Omicron mt variant BA.1 of the S protein’s RBD in complex with the ACE2 host receptor protein (7WPB) are illustrated in Figure 16b.

The binding interaction details of OEU and OC are presented in Appendix A (Omicron BA.1 and BA.2 mt S proteins’ RBD/ACE2 complex (PDB ascension N’s 7WPB and 7XO9)) of the Appendix A.

The stabilization of OEU into the ACE2 human receptor via H-bond, π-polar, polar, Hph, electrostatic, and π-alkyl contacts involves the following binding residues: L95, A99, V209, K562, E564, E208, W566, L391, A396, D206, E208, and K562.

The stabilization of OC into the ACE2 human receptor is achieved through H-bond, π-polar, polar, Hph, and π-alkyl contacts, with the following binding residues: E564, K562, A396, E208, L95, Q98, A99, and Q102. Common binding interactions between OEU and OC were revealed to be residues Glu (E564), Lys (K562), Ala (A396), Glu (E208), Leu (L95), and Ala (A99).

##### Omicron BA.2 mt S Proteins’ RBD/ACE2 Complex (Ascension Nr 7ZF7)

Binding energies for the best docking pose of OEU and OC on the protein complex between ACE2 and the mutant (mt) Omicron BA.2 mt S proteins’ RBD (PDB ID: 7ZF7) are summarized in Table 2. From Table 2, it is deduced that OEU exhibited better binding capacity compared to OC. The binding of OEU and OC on the Omicron BA.2 mt RBD/ACE2 complex is illustrated in Figure 17a.

OEU is docked in a pocket adjacent to the place occupied by the ACE2 inhibitor MLN-4760, but not sharing common contacts with it. OEU is surrounded by helices α9 and α10, the edge of the α12, α13, and α14 helices, and the loop between helices α8 and α9. On the other hand, OC is placed above the α2 helix of ACE2 in close contact to the edges of the α3, α6, and α7 helices and in proximity to the loop between α12 and α13, the edge of α19, and the loop between α18 and α19.

Two-dimensional schematic interaction models showing the binding contacts of OEU and OC with Omicron mt variants BA.2 of S protein’s RBD in complex with ACE2 host receptor protein (7ZF7) are illustrated in Figure 17b.

The binding interaction details of OEU and OC are presented in Appendix A (Omicron BA.2 mt S proteins’ RBD/ACE2 complex (ascension Nr 7ZF7)) of the Appendix A. OEU and OC make numerous contacts with the participation of H-bond, π-polar, polar, Hph, salt bridge, and π-alkyl contacts. OEU is stabilized via contacts with T276, N290, I291, M366, D367, A413, P415, T434, F438, K441, and N442. Binding interactions of OC with ACE2 were revealed to involve: L95, N98, A99, N102, D206, Y207, E208, A396, N397, K562, E564, and W566 residues.

**Figure 17 molecules-27-07572-f017:**
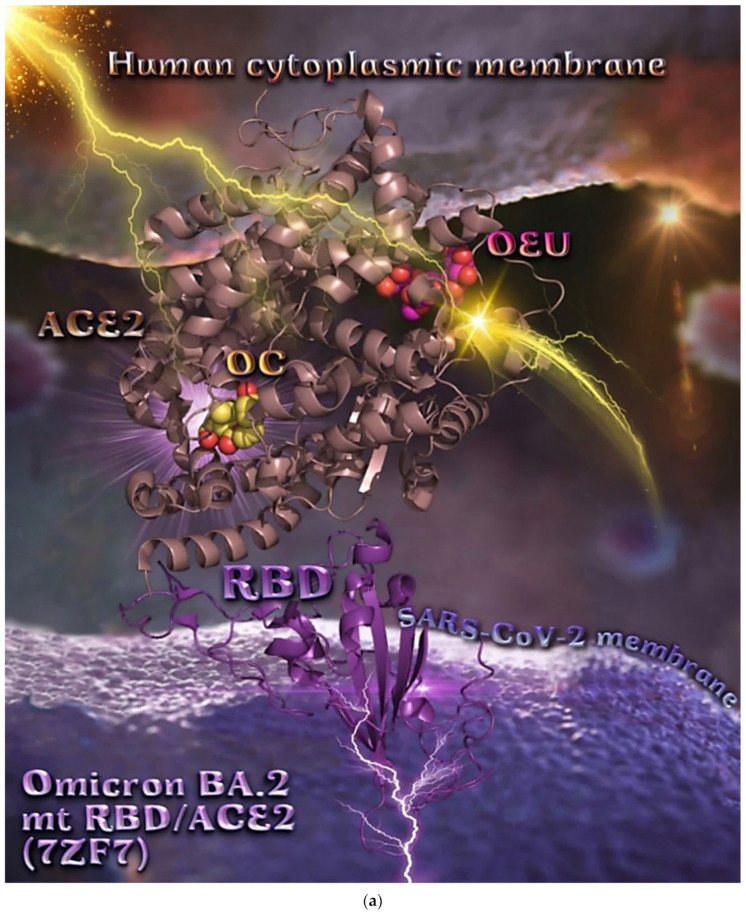
(**a**) Binding pose architecture of OEU and OC on the crystal structure of SARS-CoV-2 Omicron mt variants BA.2 of S protein’s RBD in complex with ACE2 host receptor protein (PDB ID: 7ZF7). ACE2 protein is illustrated in dark-salmon cartoon, while RBD protein is depicted in violet-purple cartoon. OEU and OC are rendered in sphere mode colored by atom type in hot-pink and yellow-orange C atoms, respectively. Hydrogen atoms are omitted from all molecules, and sugar molecules glycosylating the protein are hidden for clarity. Heteroatom color code: O—red. The final structure was ray-traced and illustrated with the aid of PyMol Molecular Graphics Systems. (**b**) Schematic 2D interaction diagrams showing the binding contacts of OEU and OC on SARS-CoV-2 Omicron mt variants BA.2 of S protein’s RBD in complex with ACE2 host receptor protein (PDB ID: 7ZF7). The final structure was illustrated with the aid of BIOVIA Discovery Studio 2016.

#### 2.3.3. Spike-Monoclonal Antibodies

##### Omicron BA.2 mt S Protein in Complex with Fab BD55-5840 (7X6A)

The binding energies for the best docking pose of OEU and OC on Omicron BA.2 mt S protein in complex with Fab BD55-5840 (cryo-EM structure PDB ID: 7X6A) are summarized in Table 1. From Table 1, it is deduced that OEU exhibited better binding capacity compared to OC. The overall structure of the Spike/Fab BD55-5840 protein complex is shown in Figure 18a. In each RBD, one Fab molecule is shown to be bound. OEU is predicted to be accommodated in the only one RBD-up of protomer C. The other two RBDs are in the “down” conformation state. The recently developed non–competing neutralizing antibodies (NAbs) cocktail, Fab BD55-5840 (also known as SA58; class 3), displayed high potency against the Omicron subvariants.

The interacting residues of the Omicron BA.2 mt variant of the SARS-CoV-2 Spike glycoprotein in complex with Fab BD55-5840 (7X6A) with OEU and OC are depicted in the 2D interaction diagrams of Figure 18b.

**Figure 18 molecules-27-07572-f018:**
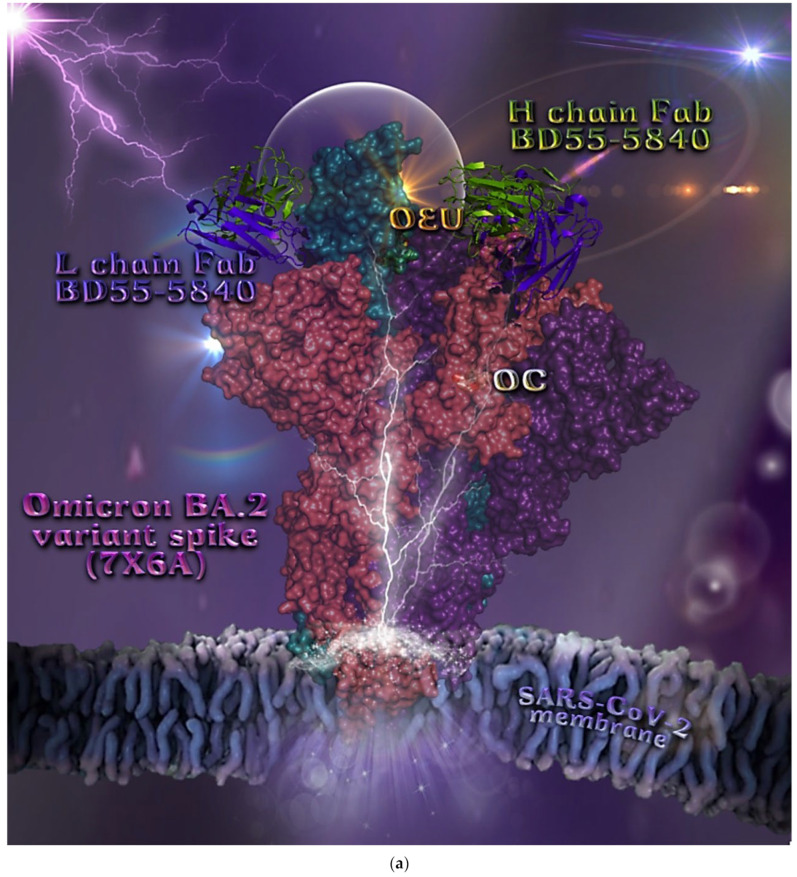
(**a**) Docking pose orientation of OEU and OC on the crystal structure of Omicron BA.2 mt variant of SARS-CoV-2 Spike glycoprotein in complex with Fab BD55-5840 (PDB: 7X6A). Spike protein is illustrated in deep-salmon, violet-purple, and deep-teal surface, for protomers A, B, and C, respectively, while the light (L) and heavy (H) chains of Fab BD55-5840 are depicted in purple-blue and split-pea-green cartoons, respectively. OEU and OC are rendered in sphere mode colored by atom type in yellow-orange and white C atoms, respectively. Only one RBD-up of protomer C is depicted inside a transparent sphere. Hydrogen atoms are omitted from all molecules, and sugar molecules glycosylating the protein are hidden for clarity. Heteroatom color code: O—red. The final structure was ray-traced and illustrated with the aid of PyMol Molecular Graphics Systems. (**b**) Schematic 2D interaction diagrams showing the binding contacts of OEU and OC on Omicron BA.2 mt variant of SARS-CoV-2 Spike glycoprotein in complex with Fab BD55-5840 (PDB: 7X6A). Solvent-accessible surfaces for each residue are depicted in light-blue spheres. The final structure was illustrated with the aid of BIOVIA Discovery Studio 2016.

#### 2.3.4. RBD-Monoclonal Antibodies

##### N501Y mt RBD in Complex with COVOX-269 Fab (7NEG)

The binding energies for the best docking pose of OEU and OC on N501Y mt RBD in complex with COVOX-269 Fab (PDB ID: 7NEG) are summarized in Table 1. From Table 1 it is deduced that OC exhibited better binding capacity compared to OEU. The overall structure of the RBD/COVOX-269 Fab protein complex is shown in Figure 19a.

The binding of OEU with the RBD is shown to be mediated at the interface between the RBD and the light (L) and heavy (H) chains of Fab-269 and adjacent to RBM in the apical position of helices α4 and α5. The interaction is mainly governed by the H-bond, and secondarily by the hydrophobic, polar and mixed π-alkyl-type hydrophobic, polar, π-polar, and π-cation, π-anion-charged electrostatic interactions. The stabilization of the OEU molecule in this protein complex is achieved mainly with its interactions with RBD. OEU does not make any contact with the Y501 mt residue (closest distance ca. 11.2 Å).

The interacting residues of the mt SARS-CoV-2 S glycoprotein’s RBD in complex with COVOX-269 Fab (PDB: 7NEG) with OEU and OC are depicted in the 2D interaction diagrams of Figure 19b.

The stabilization of OEU on the mt SARS-CoV-2 S glycoprotein’s RBD in complex with COVOX-269 Fab is achieved with the formation of the H-bond, salt bridge, Hph, polar, π-polar, π-alkyl, p-anion, and π-cation. The following binding residues of OEU with the N501Y mt RBD in complex with COVOX-269 Fab were observed: (a) with RBD: K417, R403, D405, E406, R408, Q409, and Q414; (b) with H chain of Fab-269 (hot-pink cartoon): Y52, F58, Y59, D61, K64; and (c) with L chain of Fab-269 (deep-purple cartoon): N92, Y94, P95, and A96.

**Figure 19 molecules-27-07572-f019:**
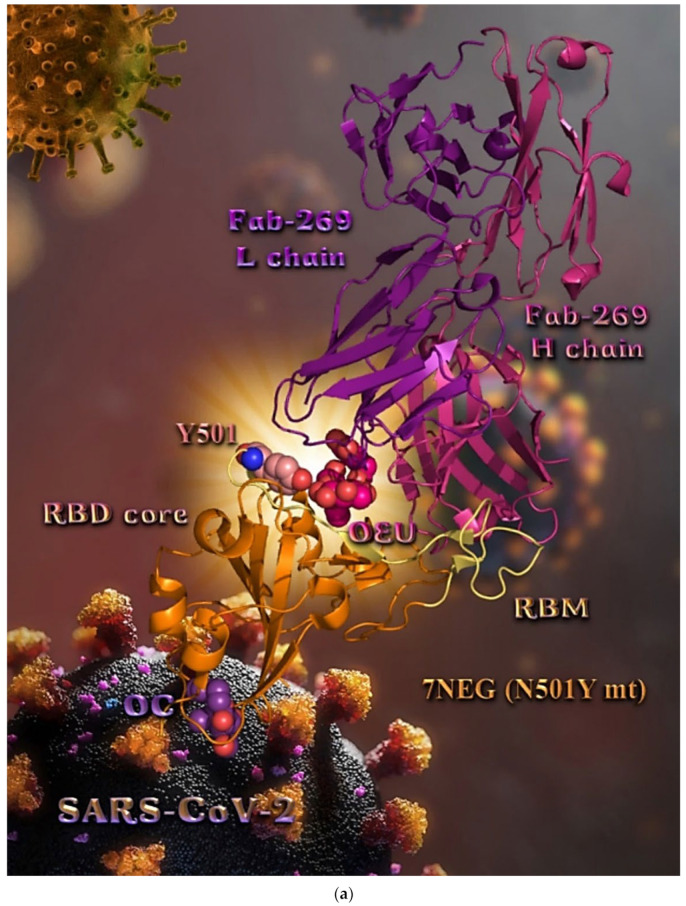
(**a**) Docking pose orientation of OEU and OC on the crystal structure of mt SARS-CoV-2 S glycoprotein’s RBD in complex with COVOX-269 Fab (PDB: 7NEG with mutation N501Y bearing Y501 mt residue). RBD (core) protein and receptor-binding motif (RBM) are illustrated as orange and yellow-orange cartoons, respectively, while the light (L) and heavy (H) chains of Fab-269 are depicted in deep-purple and hot-pink cartoon, respectively. OEU and OC are rendered in sphere mode colored by atom type in hot-pink and violet-purple C atoms, respectively. Wt Y501 residue is depicted in sphere model colored in salmon C atoms. Hydrogen atoms are omitted from all molecules, and sugar molecules glycosylating the protein are hidden for clarity. Heteroatom color code: O—red, N—blue. The final structure was ray-traced and illustrated with the aid of PyMol Molecular Graphics Systems. (**b**) Schematic 2D interaction diagrams showing the binding contacts of OEU and OC on mt SARS-CoV-2 S glycoprotein’s RBD in complex with COVOX-269 Fab (PDB: 7NEG with mutation N501Y bearing Y501 mt residue). Solvent-accessible surfaces for each residue are depicted in light-blue spheres. The final structure was illustrated with the aid of BIOVIA Discovery Studio 2016.

On the other hand, OC is stabilized at the base of the RBD protein domain, away from the binding interface between RBD and COVOX-269 Fab. OC is placed in a binding pocket flanked by the base of antiparallel β strands β1, β3, and β7 and helices α1 and α3, as well as the loop connecting the β1 strand with α3 helix (359SNCVA363 motif) and the part of the loop connecting β2 with β3 originating from the middle of β2/β3 to the end of β3 (390LCFTN394 motif) of RBD-core. The stabilization of OC may be attributed mainly to H-bonds and secondarily to hydrophobic, mixed π-alkyl-type hydrophobic, and π-polar interactions. The binding interactions of OC with RBD were revealed to be mediated via the residues: N360, C361, A363, Y365, N388, L390, F392, and V395. The N360, C361, and A363 residues belong to the loop between β1/α3; N388 to the loop between β2/β3; L390 and F392 to the part of loop connecting β2 with β3; while Y365 to the α3 helix, and V395 is the first residue of strand β3.

The binding interaction details of OEU and OC are presented in Appendix A (N501Y mt RBD in complex with COVOX-269 Fab (7NEG)) of the Appendix A.

##### Omicron BA.4-5 mt RBD in Complex with Beta-27 Fab and C1 Nanobody (7ZXU)

The binding energies for the best docking pose of OEU and OC on Omicron BA.4-5 mt RBD in complex with Beta-27 Fab and C1 nanobody (PDB ID: 7ZXU) are summarized in Table 1. From Table 1, it is deduced that OC exhibited better binding capacity compared to OEU. The overall structure of the RBD/Beta-27 Fab/C1 nanobody protein complex is shown in Figure 20a. From the binding pose, it is deduced that both EVOO constituents are shown to be accommodated exactly at the same place. Noticeably, both OEU and OC seem to be bound at the interface between RBD and both light (L) and heavy (H) chains of Beta-27 Fab. This is also the binding site of RBD on the ACE2 receptor. The interacting residues of the Omicron BA.4-5 mt SARS-CoV-2 S glycoprotein’s RBD in complex with the Beta-27 Fab and C1 nanobody (7ZXU) with OEU and OC are depicted in the 2D interaction diagrams of Figure 20b. 

### 2.4. Protein-Protein Docking Calculations on ACE2-RBD Complex

It is interesting to investigate the effect of OEU and OC to the differentiated binding of SARS-CoV-2 S protein’s RBD with human ACE2, since the driving force for the virus to infect human cells is the initial binding of the S1 subunit of the S protein followed by the S2 subunit, in particular mediating the membrane fusion [107]. The question to answer is how the binding of these compounds on either RBD or ACE2 could influence firstly the connection of RBD with ACE2 and then consequently the entry of the virus to the host cell. To this end, protein–protein docking studies were conducted using as template the wild-type 6VW1 crystal structure (Table 3).

At first, the protein–protein docking between ACE2 and RBD placed the energy basis for the remaining docking energy comparisons, and subsequently, OEU and OC were docked to their ACE2-RBD complex. Afterwards, each protein compartment was docked to the adduct of OEU or OC with the other. From Table 3, the following binding capacity order is obvious: ACE2-[RBD/OEU] > ACE2-[RBD/OC] > ACE2-RBD > RBD-[ACE2/OEU] > RBD-[ACE2/OC] > [ACE2-RBD]/OEU > [ACE2-RBD]/OC. The results revealed that the binding of either OEU or OC with ACE2 destabilizes its binding with the RBD domain of the S protein by restricting its binding capacity with RBD. The reverse (stabilization) is observed for the binding of EVOO constituents, firstly with RBD and subsequently with ACE2. It is thus concluded that most fruitful inhibition of ACE2-RBD binding may result if OEU and OC is firstly bound to ACE2 protein. It is also interesting to notice that both OEU and OC achieved better binding capacity when docked to ACE2 alone, compared to either RBD alone or the ACE2-RBD complex.

These results indicate that due to presence of OEU and OC, the bound structure of ACE2 and the Spike protein fragment becomes unstable. As a result, these phytochemical products can impart antiviral activity in SARS-CoV-2 infection.

## 3. Computational Methods

### In Silico Computational Methods (Molecular Docking Calculations)

A series of in silico studies were employed in order to predict the potential antiviral activity of the studied EVOO constituents. We performed adopted calculations by employing molecular docking, which is are a powerful tool in drug design development [108,109], on SARS-CoV-2 viral infection target proteins, including: (a) the S protein in either down (closed) or up (open) conformation state, in both wild-type and mutant S proteins; (b) the S protein in complex with host human ACE2 receptor in both wild-type and mutant S proteins; (c) the RBD domain of the S protein, either alone or in complex with ACE2 receptor, in both wild-type and mutant S proteins; and (d) the RBD of the S protein in complex with monoclonal antibodies, in both wild-type (wt) and mutant (mt) variants. Finally, we employed protein–protein docking calculations on ACE2-RBD complex.

Details concerning the computation procedures are given in the Appendix A.

## 4. Conclusions

The employed in silico calculations provided the foundation to further test the studied EVOO constituents experimentally and elucidate the role that they can play in COVID-19 treatment. Herein, the propensity of OEU and OC to act as potent SARS-CoV-2 antiviral therapeutics was explored.

The substantial capacity of OEU and OC displayed by the computation process for binding and interfering to the Spike target protein of SARS-CoV-2 responsible for the infection of the virus provided useful complementary insights for the understanding of their antiviral mechanism of action. Both OEU and OC EVOO constituents exhibited high binding affinity to the SARS-CoV-2 Spike protein, suggesting the potential utility of these compounds in the treatment of SARS-CoV-2. This study showed that the studied EVOO constituents can bind to the SARS-CoV-2 S protein blocking the interface of ACE2 and RBD-S binding, and in doing so, prevent its initial interaction with the ACE2 receptor. Since the RBD region of the SARS-CoV-2 S protein interacts with the host cell ACE2 receptor to form the RBD/ACE2 complex, which is responsible for mediation of virus invasion, OEU and OC may disrupt the interaction of ACE2 with RBD, interfering with viral entry into host cells. Additionally, binding of the EVOO constituents with other sites of S protein, apart from the RBD domain, may interfere with the substantial conformational change in the S protein and its refolding, therefore inhibiting the viral infection process. Concerning the explored currently designated mutated VOC, as well as the mutated VOI, the bioactive compounds were found to impose an impact on the role of these mutations on binding to the RBD, Spike, and RBD-ACE2 complex proteins. Together, our results suggest that the EVOO constituents inhibit the interaction of the virus with host cells through binding to the Spike protein via the RBD domain or other binding sites, ACE2, or the RBD-ACE2 protein complex.

The in silico molecular docking procedure suggests that the EVOO bioactive constituents may play a role in the therapeutic approaches in the search for pharmacological intervention on COVID-19. The findings from the docking studies provide mechanistic insights into the binding of the studied EVOO constituents on the SARS-CoV-2 spike-RBD/ACE2 protein complex, and an ample rationale for the continued study of these key components of EVOO in the fight against this lethal virus and the future exploitation of them in the development of novel therapeutic agents.

From this study, we may conclude that the above-stated active phytochemicals are predicted to have the potential to be used as anti-COVID-19 therapeutics, although the data should be validated with in vitro and in vivo tests. Future studies and experimental validation are needed for a more comprehensive understanding of the cellular networks involved in the diverse protective effects of the compounds, to elucidate the underlying mechanism of action and to ultimately develop these EVOO compounds as antiviral therapeutics against SARS-CoV-2.

## Figures and Tables

**Figure 1 molecules-27-07572-f001:**
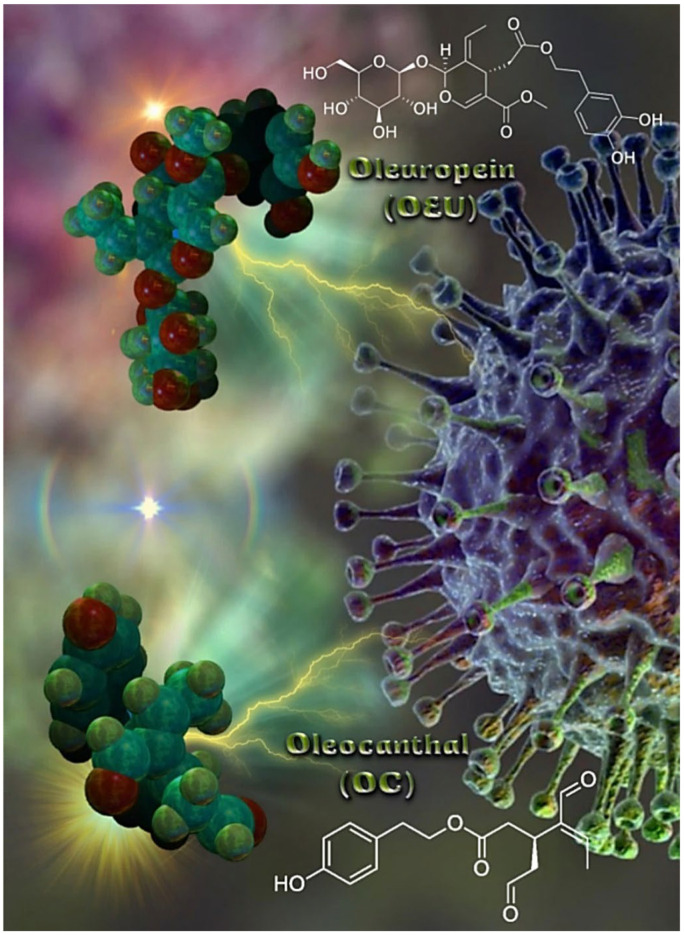
The molecular structures of EVOO constituents oleuropein (OEU) and oleocanthal (OC), generated with the aid of YASARA molecular graphics, modeling and simulation bioinformatics package v. 20.12.24 [46] in sphere representation (atom color code: C in forest green, O in firebrick red, and H in split-pea green).

**Figure 2 molecules-27-07572-f002:**
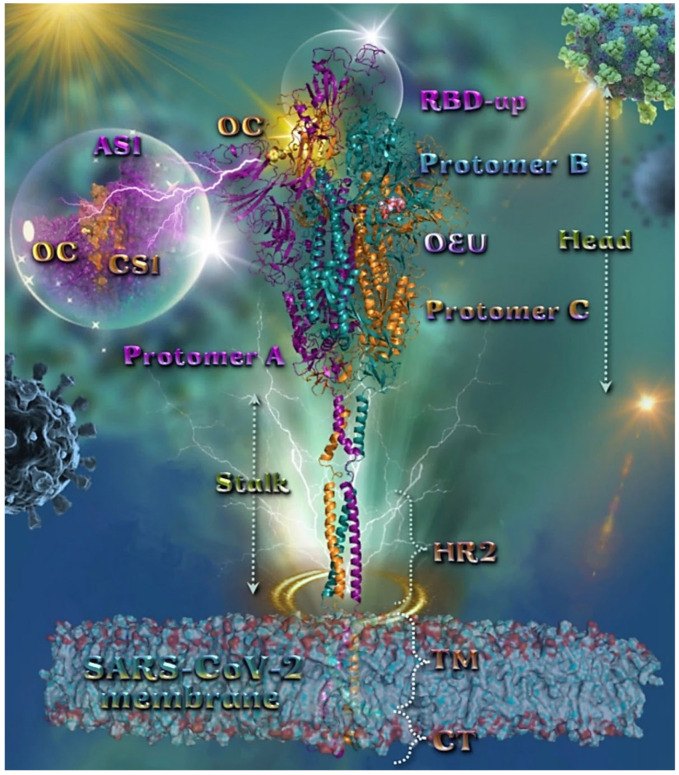
Docking pose orientation of best-bound OEU and OC molecules on the crystal structure of wild-type (wt) SARS-CoV-2 full-length model of the Spike protein in the open conformation state (one RBD-up), based on PDB: 6VSB and embedded in a realistic membrane environment of lipid bilayer mimicking the composition of the endoplasmic reticulum–Golgi intermediate compartment after molecular dynamics simulation [75]. All structural models were downloaded from the Amaro lab (https://amarolab.ucsd.edu/covid19.php (accessed on 13 June 2021)). Target trimeric wt S protein is illustrated as cartoon colored by chain in deep purple, deep teal, and orange for each of the 3 protomers (a, b, and c, respectively). The critical one RBD-up domain is also indicated inside a transparent sphere. OEU and OC are rendered in sphere mode and colored according to atom type in light-pink and yellow-orange C atoms, respectively. OC is stabilized at the interface between the NTD (part of the S1) AS1 of protomer a and the RBD domain (CS1) of protomer c (inlaid depiction in transparent sphere). OEU is stabilized at the interface between the C-terminal domain 1 (CT1) of protomer b and the HR1 and CH domains of protomer c, being at the apical position of both the fusion peptide (FP) and fusion peptide region (FPR) of protomer c, flanked by the central helix (CH). In the structure are also depicted the heptad repeat 2 (HR2, 1163–1210), the transmembrane domain (TM, 1214–1234), and the cytoplasmic tail (CT, 1235–1273) domains. Color code used for lipid tails (surface representation): POPC, POPE, POPI, POPS, and cholesterol in cyan. P atoms of the lipid heads and cholesterol’s O3 atoms are highlighted in red. N-linked glycans (NAG moieties) are omitted from the structure for clarity. Molecular docking simulations were performed individually. Hydrogen atoms are omitted from both molecules, and sugar molecules glycosylating the protein are hidden for clarity. Heteroatom color code: O—red. The final structure was ray-traced and illustrated with the aid of PyMol Molecular Graphics Systems.

**Figure 4 molecules-27-07572-f004:**
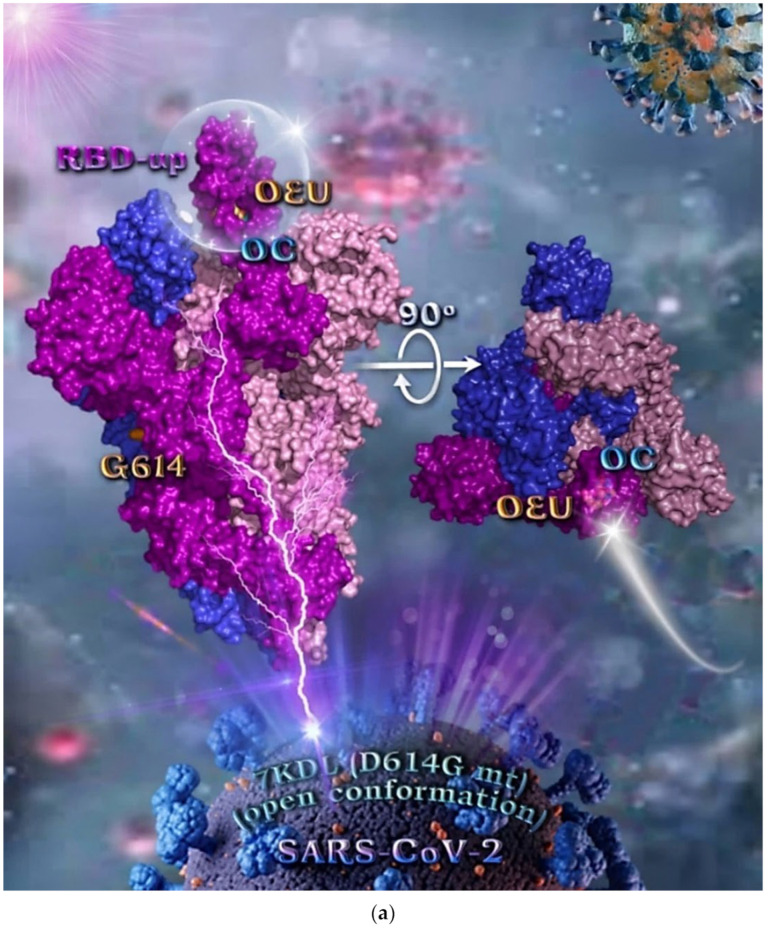
(**a**) Docking pose orientation of best-bound OEU and OC molecules on the crystal structure of mutant (mt) variant of SARS-CoV-2 S protein trimer bearing the D614G mutation in which aspartic acid (D) residue in 614 position is substituted by a glycine (G) residue (PDB entry code 7KDL). The protein is illustrated in its open conformation state with one RBD in up position. The critical one RBD-up domain is also indicated inside a transparent sphere. The trimeric target protein is illustrated as opaque surface with subdomains color-coded according to chain (protomers a, b, and c or chains A, B, and C in tv blue, purple, and light-pink color, respectively) with additional depiction of Gly614 (G614) mutated residue highlighted in yellow-orange. A view of the docking pose derived by a 90-degree rotation of the structure around the S1/S2 furin cleavage site (at the G614 site) is also indicated. OEU and OC molecules (docked independently) are rendered in sphere mode and colored according to atom type in yellow-orange and cyan C atoms, respectively. Hydrogen atoms are omitted from both molecules, and sugar molecules glycosylating the protein are hidden for clarity. Heteroatom color code: O—red. The final structure was ray-traced and illustrated with the aid of PyMol Molecular Graphics Systems. (**b**) Schematic 2D and 3D interaction diagrams showing the binding contacts of OEU and OC on 7KDL S protein. Residues rendered in either sphere or stick model are colored by interaction type and slate blue for 2D and 3D diagrams, respectively. OEU and OC rendered in line and ball-and-stick model (2D and 3D, respectively) are colored by atom type in grey and brown C atoms, respectively. Interactions are depicted in dotted lines colored according to interaction type. Solvent-accessible surfaces for each residue in 2D diagrams are depicted in light-blue spheres surrounding the residue spheres. The final structure was illustrated with the aid of BIOVIA Discovery Studio 2016.

**Figure 5 molecules-27-07572-f005:**
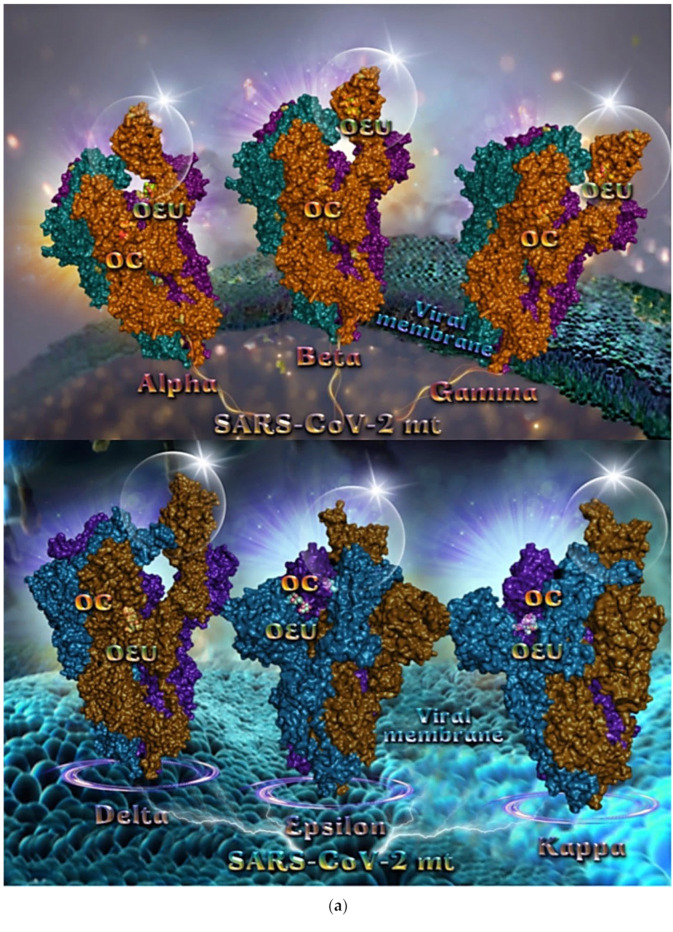
(**a**) Docking pose orientation of best-bound OEU and OC molecules on the crystal structure of mutant (mt) variant of SARS-CoV-2 S trimeric proteins Alpha (PDB ID: 8DLI), Beta (PDB ID: 8DLL), Gamma (PDB ID: 8DLO) (**upper panel**), Delta (PDB ID: 7V7O), Epsilon (PDB ID: 8DLT), and Kappa (PDB ID: 7V7E) (**lower panel**). The proteins are illustrated in their open conformation state with one RBD in up position. The critical one RBD-up domains are also indicated inside transparent spheres. The trimeric target proteins are illustrated as semitransparent surfaces with subdomains color-coded according to chain (protomers a, b, and c or chains A, B, and C in deep purple, deep teal, and orange, respectively, in upper part and purple blue, deep sky blue, and deep brown, respectively, in lower part). OEU and OC molecules (docked independently) rendered in sphere mode and colored according to atom type in split-pea green and yellow-orange C atoms, respectively. Hydrogen atoms are omitted from both molecules, and sugar molecules glycosylating the protein are hidden for clarity. Heteroatom color code: O—red. The final structure was ray-traced and illustrated with the aid of PyMol Molecular Graphics Systems. (**b**) A schematic 2D interaction diagrams showing the binding contacts of OEU and OC on Delta (PDB ID: 7V7O) S protein. Solvent-accessible surfaces for each residue are depicted in light-blue spheres. The final structure was illustrated with the aid of BIOVIA Discovery Studio 2016.

**Figure 6 molecules-27-07572-f006:**
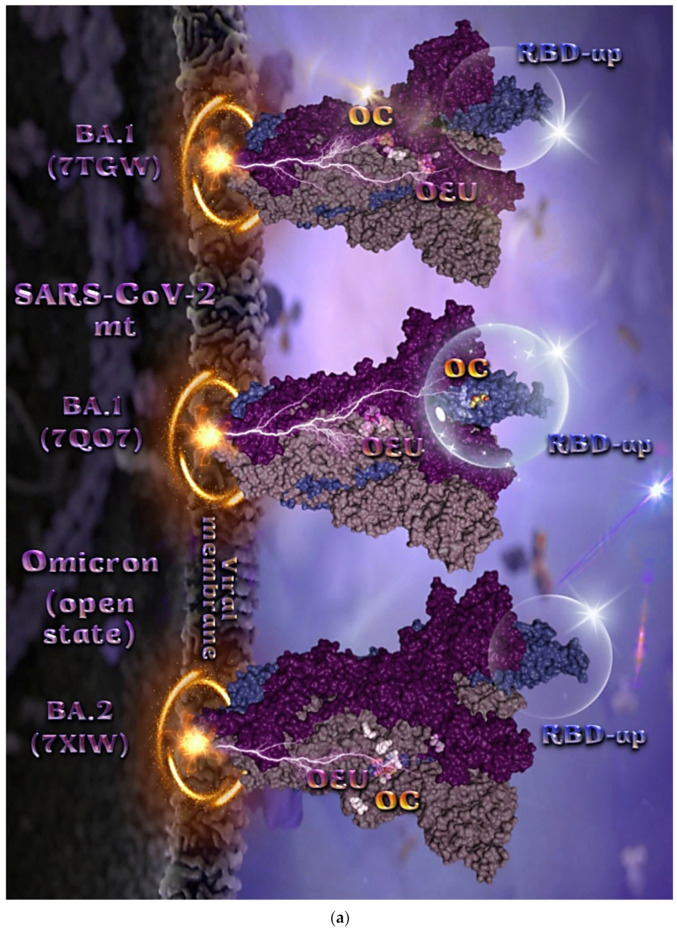
(**a**) Docking pose orientation of best-bound OEU and OC molecules, on the crystal structure of mutant (mt) Omicron BA.1 (PDB IDs: 7TGW, 7QO7), BA.2 (PDB ID: 7XIW) variants of SARS-CoV-2 S trimeric proteins in open conformation state with one RBD in up position. The critical one RBD-up domains are also indicated inside transparent spheres. The trimeric target proteins are illustrated as semitransparent surfaces with subdomains color-coded according to chain (protomers a, b, and c or chains A, B, and C in dirty violet, deep purple, and slate blue, respectively). OEU and OC molecules (docked independently) rendered in sphere mode and colored according to atom type in violet and yellow-orange C atoms, respectively. Hydrogen atoms are omitted from both molecules and sugar molecules glycosylating the protein are hidden for clarity. Heteroatom color code: O—red. The final structure was ray-traced and illustrated with the aid of PyMol Molecular Graphics Systems. (**b**) A schematic 2D interaction diagrams showing the binding contacts of OEU and OC on Omicron BA.1 (PDB IDs: 7QO7) S protein. Solvent-accessible surfaces for each residue are depicted in light-blue spheres. The final structure was illustrated with the aid of BIOVIA Discovery Studio 2016.

**Figure 8 molecules-27-07572-f008:**
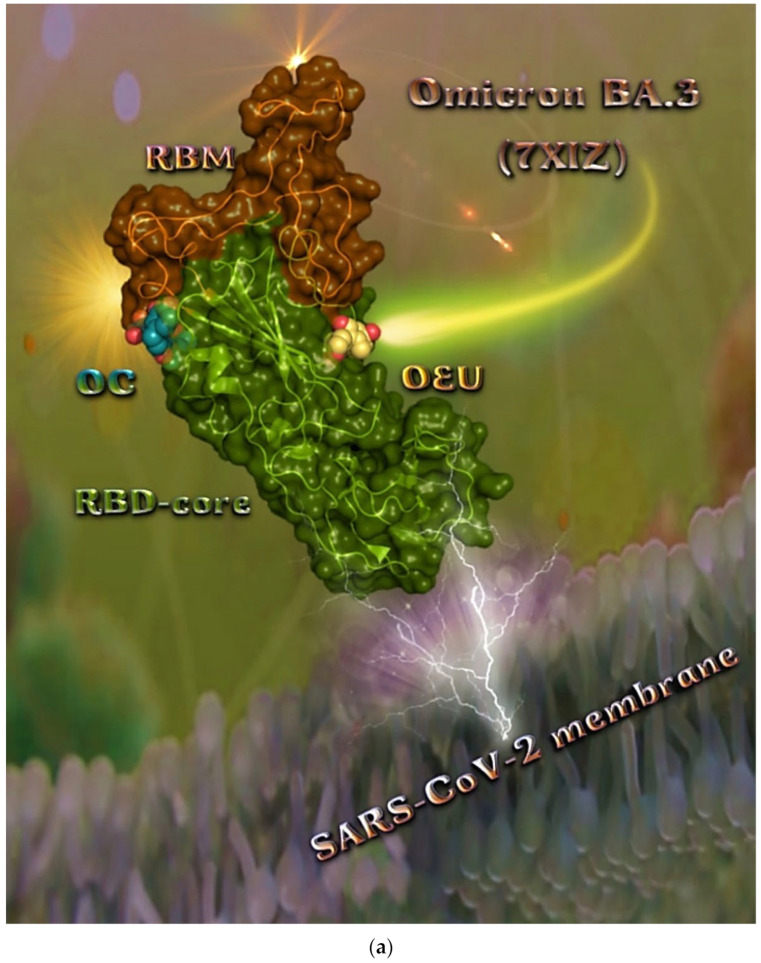
(**a**) Docking pose architecture of best-bound OEU and OC molecules on the crystal structure of mutant (mt) Omicron BA.3 variant of SARS-CoV-2 Spike trimeric glycoprotein’s RBD (PDB ID: 7XIZ). RBD (core) protein and RBM motif are illustrated as deep split-pea-green and deep-orange cartoons, respectively, with additional depiction of semitransparent surface also colored by the cartoon colors. Both compounds are rendered in sphere mode colored by atom type in yellow-orange and deep teal for OEU and OC, respectively. Hydrogen atoms are omitted from all molecules and sugar molecules glycosylating the protein are hidden for clarity. Heteroatom color code: O—red. The final structure was ray-traced and illustrated with the aid of PyMol Molecular Graphics Systems. (**b**) Schematic 2D interaction diagrams showing the binding contacts of OEU and OC on Omicron BA.3 variant of SARS-CoV-2 Spike trimeric glycoprotein’s RBD (PDB ID: 7XIZ). Solvent-accessible surfaces for each residue are depicted in light-blue spheres. The final structure was illustrated with the aid of BIOVIA Discovery Studio 2016.

**Figure 11 molecules-27-07572-f011:**
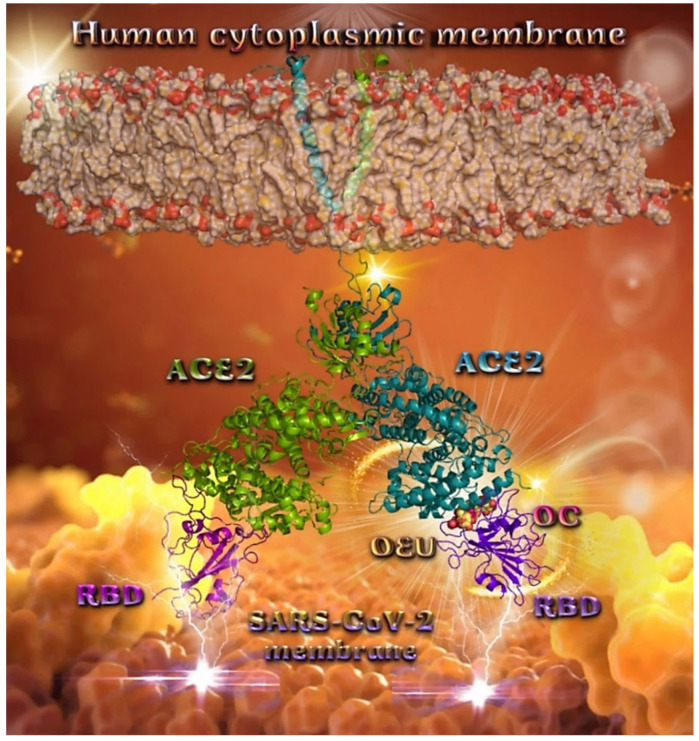
Docking pose orientation of best-bound OEU and OC molecules on the crystal structure of wild-type (wt) SARS-CoV-2 full-length S protein’s RBD bound to the ACE2 receptor (from PDB ID: 6M17). The ACE2 receptor is depicted in its dimer structure with chains A and B with additional illustration of the transmembrane and extracellular domain. ACE2 receptor is shown to be embedded in a realistic membrane environment of lipid bilayer mimicking the human cytoplasmic membrane after molecular dynamics simulation. All structural models were downloaded from the Amaro lab (https://amarolab.ucsd.edu/covid19.php (accessed on 13 June 2021)). The protein complex is illustrated as cartoon colored by chain in deep purple for the RBD and deep teal for one monomer of the dimeric ACE2 receptor. OEU and OC molecules are rendered in sphere mode and colored according to atom type in yellow-orange and hot-pink C atoms, respectively. Both OEU and OC molecules are shown to be anchored in the interface between the RBD and ACE2 proteins. Color code used for lipid tails (surface representation): POPC, POPE, POPI, POPS, and cholesterol in wheat. P atoms of the lipid heads and cholesterol’s O3 atoms are highlighted in red. Molecular docking simulations were performed individually. Hydrogen atoms are omitted from both molecules for clarity. Heteroatom color code: O—red. The final structure was ray-traced and illustrated with the aid of PyMol Molecular Graphics Systems.

**Figure 16 molecules-27-07572-f016:**
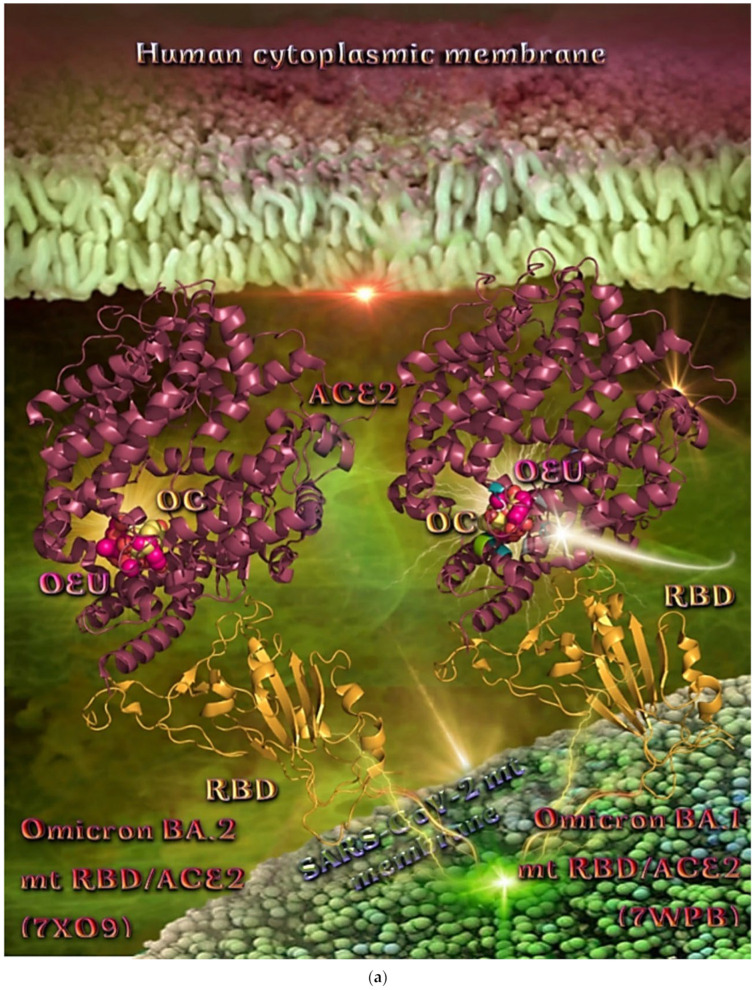
(**a**) Binding pose architecture of OEU and OC on the crystal structure of SARS-CoV-2 Omicron mt variants BA.1 and BA.2 of S protein’s RBD in complex with ACE2 host receptor protein (PDB IDs: 7WPB and 7XO9, respectively). ACE2 protein is illustrated in deep-salmon cartoon, while RBD protein is depicted in orange cartoon. OEU and OC are rendered in sphere mode colored by atom type in hot pink and yellow-orange, respectively. Selected binding residues of OEU and OC on Omicron BA.1 variant are highlighted in the cartoon. Hydrogen atoms are omitted from all molecules, and sugar molecules glycosylating the protein are hidden for clarity. Heteroatom color code: O—red. The final structure was ray-traced and illustrated with the aid of PyMol Molecular Graphics Systems. (**b**) Schematic 2D interaction diagrams showing the binding contacts of OEU and OC on SARS-CoV-2 Omicron mt variant BA.1 of S protein’s RBD in complex with ACE2 host receptor protein (PDB IDs: 7WPB). The final structure was illustrated with the aid of BIOVIA Discovery Studio 2016.

**Figure 20 molecules-27-07572-f020:**
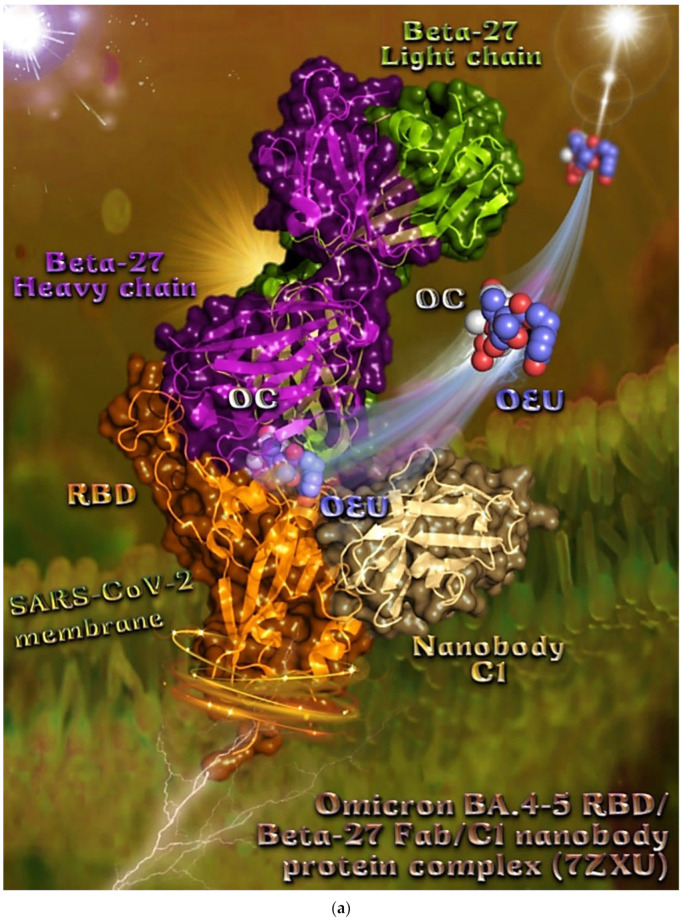
(**a**) Docking pose orientation of OEU and OC on the crystal structure of Omicron BA.4-5 mt SARS-CoV-2 S glycoprotein’s RBD in complex with Beta-27 Fab and C1 nanobody (PDB: 7ZXU). RBD and C1 nanobody are illustrated as orange and yellow-orange cartoons, respectively, while the light (L) and heavy (H) chains of Beta-27 Fab are depicted in split-pea-green and deep-purple cartoon, respectively, with additional depiction, for all protein structures, of semitransparent surface colored according to cartoon colors. OEU and OC are rendered in sphere mode colored by atom type in slate-blue and white C atoms, respectively. Hydrogen atoms are omitted from all molecules for clarity. Heteroatom color code: O—red. The final structure was ray-traced and illustrated with the aid of PyMol Molecular Graphics Systems. (**b**) Schematic 2D interaction diagrams showing the binding contacts of OEU and OC on Omicron BA.4-5 mt SARS-CoV-2 S glycoprotein’s RBD in complex with Beta-27 Fab and C1 nanobody (PDB: 7ZXU). Solvent-accessible surfaces for each residue are depicted in light-blue spheres. The final structure was illustrated with the aid of BIOVIA Discovery Studio 2016.

**Table 3 molecules-27-07572-t003:** ΔG_bind_ glide extra precision (XP) binding energies (in kcal/mol) of EVOO constituents OC and OEU docked firstly on either RBD or ACE2, and consequently ACE2 or RBD, respectively, or on their protein–protein complex ACE2-RBD and ACE2 and RBD alone. The binding energy of the reference protein–protein docking between ACE2-RBD is also reported for comparison (template PDB ID: 6VW1).

Protein-Protein/OEU, OC Complexes	Binding Energies (kcal/mol)
ACE2-RBD	−69.63
[ACE2-RBD]/OEU	−37.93 (from Table 2)
[ACE2-RBD]/OC	−29.48 (from Table 2)
ACE2-[RBD/OEU]	−72.85
RBD-[ACE2/OEU]	−67.80
ACE2-[RBD/OC]	−71.45
RBD-[ACE2/OC]	−66.11
RBD/OEU	−36.71
ACE2/OEU	−40.57
RBD/OC	−33.27
ACE2/OC	−40.37

## Data Availability

Not applicable.

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
