# Peer review of "In Silico Approach for the Evaluation of the Potential Antiviral Activity of Extra Virgin Olive Oil (EVOO) Bioactive Constituents Oleuropein and Oleocanthal on Spike Therapeutic Drug Target of SARS-CoV-2"

_molecules, 2022, doi:10.3390/molecules27217572_

Round 1
Reviewer 1 Report
The authors Elena et al. describes the ability of bioactive constituents of EVOO to serve as potent SARS-cov2 compounds. Their docking results shows strong binding to the spike protein. Overall, in silico calculations shows promising treatment and further experimental bioassay are required.
Is protein flexible in docking?
one more minor comment. I would request authors to show some of the key molecular interaction happening in RBD.
Reviewer 2 Report
The paper entitled “In silico approach for the evaluation of the potential antiviral activity of extra virgin olive oil (EVOO) bioactive constituents oleuropein and oleocanthal on spike therapeutic drug target of SARS-CoV-2” by Elena G. Geromichalou and George D. Geromichalos provided a vast amount of data in 44 pages! The authors explored a fundamental problem nowadays, namely the research into potential inhibitors of coronavirus replication. They chose two components of the olive oil – oleuropein (OEU) and oleocanthal (OC) – as prospective ligands for the virus proteins. These compounds are from different organic classes but share some structural fragments. As the coronavirus mutates quite fast, the authors investigated theoretically interactions between OC as well as OEU and several mutations of this virus, including the latest omicron variant. The studies resulted concluded that these two compounds might constitute lead compounds for the future development of potential antiviral drugs. However, I am not convinced that these two compounds can be used as medicines acquired from olive oil as their content is relatively low in this natural source.
The manuscript is extensive and rich in illustrations. However, in my opinion, several weaknesses listed below affect the strength of the paper and need to be addressed by the authors.
11. The figures are rich in colours. Nevertheless, the details concerning the docking pose are barely legible in some of the figures. I would suggest enlarging fragments of the protein surfaces to which OEU and OC are linked.
22. The authors list many interactions between the virus proteins and OEU/OC. Some of these interactions seem strong and vital, but some are rather weak. I would recommend presenting these interactions in a different way. A 2D graphics such as Ligplot diagrams could be used rather than the litany of contacts. Such graphics is more informative concerning the interactions ligand-protein than the list. Moreover, why there is inconsistency in the description of contacts – some of them are fully characterized (distance and type, e.g. lines 378-387), some are given only the distance (e.g. lines 293-308), some are devoid of any description (e.g. lines 326-339). The authors should also use uniform abbreviations for the amino acids, not once three-letter acronyms, another time one-letter symbols.
33. Another problem that would need solving is the presence of chiral centres in both molecules, two in OEU (I am not referring to the sugar component) and one in OC. How this chirality affects the energy of binding? Is there any difference between the natural compounds and their enantiomers? It is an important question for the future development of OC and OEU analogues.
44. The text is somewhat readable, but in some parts, the English language is not up to standards and needs corrections. I have found several long and ambiguous sentences, e.g. lines 247-252, 343-345, 844-845, that would benefit from splitting and clarification. There are also some spelling mistakes and wordiness that should be amended, e.g. “for shake of clarity” should be “ for the sake of clarity”, but even better “for clarity”; “in close proximity” can be shortened to “in proximity”.
Reviewer 3 Report
This paper reported a docking study about constituents of Extra Virgin Olive Oil with the Spike protein of SARS-CoV-2. The molecular docking calculations were performed with Schrödinger modeling suite. The authors compared the docking modes between different states, strains and formations of Spike proteins, and they discussed their potentials in inhibiting SARS-CoV-2. This work is interesting, however, the results are not validated by any experiments. Here are my concerns.
1) The molecular docking were performed with Schrödinger modeling suite. How about the success rate of this docking program? Can it result in acceptable prediction?
2) The binding conformation of the docking results are not clear. The figures did not illustrate the detatils of the interaction betweenatoms.
3) Are the binding energies in Table 1 and Table 2 statistically meaningful? Did the -29.48 indicate any protein-ligand binding?
4) Why did the authors select OEU and OC, but not the other constituents in Extra Virgin Olive Oil ?
5) Please provide the computational flow chart of the work.
Round 2
Reviewer 2 Report
After the corrections, I believe that the manuscript can be published in the journal.
Reviewer 3 Report
The authors have addressed many of my concerns. However, it is highly risky to discover any molecular interaction only using computational docking methods. I urge the authors try to perform some experiments to validate the discovery.